# Performance evaluation of GPU-based parallel sorting algorithms

**Mohammed Alaa Ala'anzy**[1]*, **Nurdaulet Tolendi**[1],
**Baizhan Baubek**[1], **Abdulmohsen Algarni**[2]

**1** Department of Computer Science, SDU University, Kaskelen, Kazakhstan, **2** Department of Computer Science, King Khalid University, Abha, Saudi Arabia

* m.alanzy@ieee.org

## Abstract

Sorting can be approached in two main ways: sequentially and in parallel. In sequential sorting, data is processed in a single-threaded manner, which can be slow for large datasets. However, parallel sorting divides the task across multiple processing units, enabling faster results by processing data simultaneously. Furthermore, Compute Unified Device Architecture (CUDA) technology enables developers to leverage GPU power for general-purpose parallel computing, significantly accelerating tasks like sorting. This paper investigates the GPU-based parallelization of merge sort (MS), quick sort (QS), bubble sort (BS), radix top-k selection sort (RS), and slow sort (SS) presenting optimized algorithms designed for efficient sorting of large datasets using modern GPUs. The primary objective is to evaluate the performance of these algorithms on GPUs utilizing CUDA, with a focus on analyzing both parallel time complexity and space complexity across various data types. Experiments are conducted on four dataset scenarios: randomly generated data, reverse-sorted data, already-sorted data, and nearly-sorted data. Also, the performance of GPU-accelerated implementations is compared with their sequential counterparts to assess improvements in computational efficiency and scalability. Earlier GPU-based generations of this type typically achieved acceleration rates between 2× and 9× over scalar CPU code. With newer GPU enhancements, including parallel-aware primitives and radix- or merge-optimized operations, acceleration rates have seen significant improvement. Our experiments indicate that Radix Sort based on GPUs achieves a significant speedup of approximately 50× (sequential: 240.8 ms, parallel: 4.83 ms) on 10 million random sort elements. Quick Sort and Merge Sort have 97× and 103× speedups, respectively (Quick: 1461.97 ms vs. 15.1 ms; Merge: 2212.33 ms vs. 21.4 ms). Bubble Sort, while significantly improving in parallel (123,321.9 ms to 7377.8 ms for an ≈17× improvement), is considerably worse overall. Slow Sort demonstrates a moderate but consistent acceleration, reducing execution time from 74.07 ms in the sequential version to 3.99 ms on the GPU, yielding an ≈18.6×

**Data availability statement:** All relevant data are publicly available from the Figshare repository: https://doi.org/10.6084/m9.figshare.29558357.

**Funding:** This research was financially supported by the Deanship of Scientific Research and Graduate Studies at King Khalid University under research grant number (R.G.P.2/12/46). The funders were involved in supervising the research and revising the manuscript.

**Competing interests:** The authors have declared that no competing interests exist.

speedup. These experimental findings confirm that the new single-GPU implementations can get speedups ranging from 17× to over 100×, surpassing the typical gains reported in previous generations and comparable to or over rates of acceleration reported for cutting-edge parallel sorting algorithms in recent studies.

## Introduction

The exponential growth of data-intensive fields such as artificial intelligence, scientific computing, real-time analytics, and large-scale simulations has placed unprecedented demands on algorithmic performance and scalability [1–5]. Sorting underpins many of these applications, from database query processing and graph analytics to genome sequencing and large-scale scientific simulations. Notably, sorting operations can account for up to 25% of processing time in data-intensive applications, underscoring their critical role in high-performance computing workloads [6]. As dataset sizes grow into the millions and beyond, traditional sequential sorting algorithms become performance bottlenecks, leading to unacceptable latency and resource usage.

Parallel sorting techniques on multi-core CPUs and distributed clusters have alleviated some of these pressures, but they still face challenges in load balancing, memory bandwidth, and inter-node communication overhead. In particular, multi-GPU strategies have emerged as a promising solution for overcoming the memory and computational limits of single-device sorting. Ilic et al. [7] demonstrated up to a 14× improvement in throughput by exploiting NVLink and PCIe 4.0 interconnects for distributed merge-based sorting, while Tolovski et al. [8] introduced RMG Sort, a radix-partitioning approach that segments most significant bits across multiple GPUs to achieve near-linear scalability with minimal inter-device communication.

GPUs with thousands of cores and high-bandwidth memory subsystems have become a cornerstone of parallel computing [9]. NVIDIA's CUDA programming model provides fine-grained control over thread hierarchies, shared memory, and warp-synchronous execution, enabling optimized sorting kernels that deliver substantial speedups over CPU baselines [10,11]. Indeed, recent work shows that well-tuned GPU implementations of comparison-based sorts such as MS and QS can achieve up to 14×–38× acceleration on million-element datasets [12].

Recent hardware trends in unified-memory systems have changed the CPU-versus-GPU sorting landscape significantly. For example, a study by Liu [13] evaluated Monte Carlo particle-sorting algorithms on Apple's M2 Max chip, which uses unified CPU–GPU memory. Interestingly, CPU-based sorting outperformed GPU-based sorting for partially sorted datasets, due to reduced thread divergence and the absence of off-chip transfer overhead, while GPUs still excelled on fully random data. This demonstrates the necessity of including diverse data distribution patterns when benchmarking sorting algorithms in modern heterogeneous environments.

Over the past five years, a variety of GPU-centric sorting approaches have been proposed. Radix-based methods, including hardware-accelerated radix top-K selection [14], leverage digit-wise bucketing to achieve near-linear time complexity

and excellent coalesced memory access. Segmented sorting evaluations, such as those by Schmid and Caceres [15] and Schmid et al. [16], compare fix-sort and merge-based variants across multiple GPU generations, providing recommendation maps for optimal strategy selection. Specialized order-statistics kernels like "bucketMultiSelect" [17] extract multiple percentiles more than 8× faster than full sorts, highlighting opportunities when only partial ordering is needed.

Despite these advances, existing studies often focus on one algorithmic paradigm or use disparate experimental conditions, which hinders direct comparison. Furthermore, many GPU sorting kernels rely on advanced memory optimizations, shared memory tiling, warp-level intrinsics, and manual prefetching, yet the relative benefits of these techniques across different algorithms and data distributions remain underexplored. Including simpler algorithms such as BS, despite its $O(n^2)$ complexity, provides a useful baseline for examining synchronization and memory-access behaviors under parallel execution [18]. By evaluating both naïve and sophisticated algorithms side-by-side, we gain deeper insight into how algorithmic complexity and GPU optimizations interact in practice.

To address these gaps, we implement and evaluate four representative sorting algorithms MS, QS, BS, RS adn SS, in a unified CUDA C++ framework. MS and QS represent efficient, comparison-based divide-and-conquer paradigms; BS serves as a synchronization-focused baseline; RS exemplifies digit-based, non-comparison sorting optimized for large integer datasets and SS represents a bitmap based presence encoding approach designed for efficient integer sorting with compact auxiliary memory usage. We systematically benchmark each algorithm on an NVIDIA GTX 1660 SUPER using a fixed dataset of ten million integers under four input distributions (random, sorted, reverse-sorted, and nearly sorted). By measuring execution time, memory usage, and comparing GPU implementations against C++ based sequential baselines, we quantify speedups ranging from 7× to 38× and analyze algorithm robustness and scalability.

To improve the clarity of the workflow and provide readers with an immediate understanding of the study's scope, we introduce an overview diagram (Fig 1) that summarizes the full evaluation pipeline. This schematic illustrates the dataset preparation, CPU and GPU sorting implementations, and the performance assessment components of the proposed framework.

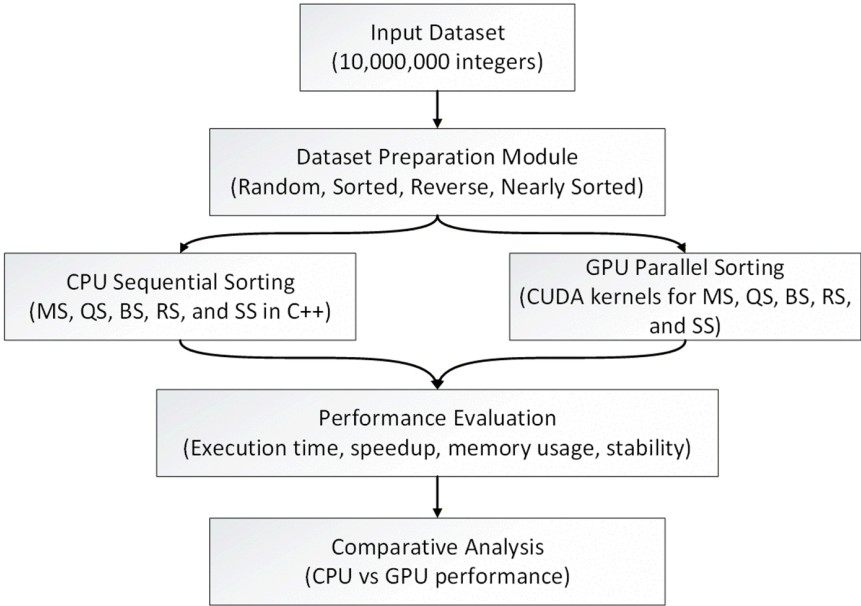

**Fig 1. Overall workflow of the dataset generation, CPU and GPU sorting algorithms, and performance comparison.**

The principal contributions of this work are as follows:

- Developed GPU-accelerated implementations of four sorting algorithms, MS, QS, BS, RS and SS, using the CUDA programming model to exploit parallel execution capabilities.
- Provided a comparative analysis of these algorithms, representing different algorithmic paradigms: divide-and-conquer (MS, QS), elementary quadratic sort (BS), digit-based non-comparison sorting (RS) and bitmap based sorting (SS).
- Evaluated each implementation across four data distribution scenarios: random, sorted, reverse-sorted, and nearly sorted, using a dataset of ten million integers to assess scalability and algorithm robustness under varying input characteristics.
- Conducted a detailed comparison between GPU-based and sequential CPU implementations to measure performance improvements in terms of execution time, speedup, and memory usage, thereby demonstrating the computational benefits of GPU acceleration.

These contributions establish a unified benchmarking framework for evaluating GPU-based sorting algorithms under consistent experimental conditions. By analyzing diverse algorithmic strategies and input scenarios, this work contributes new insights into the practical performance characteristics of classical and modern sorting algorithms when optimized for parallel execution on CUDA-enabled platforms.

## Related work

Various works have concentrated on the implementation and optimization of parallel sorting on GPUs. Ishan Joshi and Singh et al. [19] conducted a survey of several sorting algorithms running on a GPU and underlined that the most popular sorting algorithms in CUDA-based implementations are MS, QS, RS, and Sample Sort. Their work then emphasizes that although RS can be the fastest for large inputs because of its linear time complexities, comparison-based sorting methods, such as MS and QS, provide more stability in their performance with different input distributions.

Among sorting algorithms, MS has been in the limelight for a very long time regarding its suitability for parallel execution. Tanasic et al. [20] explored multi-GPU-based comparison sorting and found that cooperation by multiple GPUs yielded a substantial gain in sorting large-scale data. They were able to optimize the two most enigmatic parts, which are the partitioning of data and inter-GPU communication. Building upon these foundational ideas, Çetin et al. [21] proposed a memory-aware multi-GPU sorting framework that dynamically balances workload distribution while minimizing data transfer overhead through unified memory access and NCCL-based communication. Their evaluation on various large-scale datasets demonstrated not only improved throughput but also higher resource utilization across heterogeneous GPU configurations, making their approach more adaptable to modern CUDA environments.

While MS benefited from its structured partitioning, QS presents a very different challenge due to its heavy reliance on partitioning and recursive calls. Kumari and Singh [22] proposed a binary search-based parallel QS implementation that tried to reduce inter-thread communication overhead for partitioning efficiency.

As BS would not be used for any high-performance computing environment, some studies have developed a GPU implementation of the same for benchmarking and educational purposes. In the comparison study by Faujdar and Ghrera [18], although its performance scaled rather poorly for a large set of data, it could be useful in evaluating thread synchronization strategies and memory access optimizations in parallel architectures.

Empirical comparisons of different sorting algorithms have also been done extensively to understand their real performance on GPUs. Schmid and Caceres [16] proposed the fix sort optimization, which will enable off-the-shelf sorting algorithms to efficiently handle segmented sorting. Even though the work focuses mainly on segmented sorting, the methodology of benchmarking is very useful for general insights into parallel sorting efficiency.

Schmid et al. [15] conducted a detailed evaluation of segmented sorting strategies on GPUs, benchmarking multiple algorithms across seven devices with varied segment sizes and distributions. Their use of S-curve and heat map analyses

led to a recommendation map that selected the optimal strategy in 47.57% of cases, with a slowdown of less than 1.5× in over 93%. This study highlights the importance of input-specific benchmarking in segmented sorting performance.

Outside these algorithm-specific optimizations, other works have focused on the hardware and CUDA-based performance optimizations for sorting. Yoshida et al. [11] present the influence that different CUDA versions may have on sorting performance on GPUs. They show how warp-scheduling, pipelining instructions, and adapting to memory hierarchy changes can seriously affect execution time. These are indications that software optimizations in a CUDA-based algorithm play a much more important role in sorting than ever before, for the high throughput of modern GPUs.

Li et al. [14] solve the issue of top-k selection, finding the k largest or smallest values in a set, of utmost importance in data-intensive computing. The traditional methods utilize fast but size-constrained on-chip memory, excluding scalability with an upper bound on k. Furthermore, Li et al. [14] present a parallel and optimized radix top-k selection on GPUs scalable to much larger k values with no loss of efficiency, input batch size, and length insensitive. They utilize a novel optimization framework constructed for high resource and memory bandwidth utilization, permitting large speedup over current techniques.

Blanchard et al. [17] introduced 'bucketMultiSelect', a GPU-based algorithm for extracting multiple order statistics, such as percentiles, without fully sorting the data. For large vectors containing up to $2^{28}$ double-precision values, their method selects 101 percentile values in under 95 ms, achieving over 8× speedup compared to an optimized CUDA-based merge sort. This approach is particularly relevant in applications where specific statistics are needed but full sorting is unnecessary, offering substantial performance savings.

To further improve sorting efficiency on the GPU, researchers have explored hybrid sorting techniques that combine multiple approaches. Gowanlock and Karsin [23] suggested a hybrid CPU/GPU sorting strategy, which performs dynamic workload distribution between CPU and GPU cores for the best execution time. The approach of these authors proves how task offloading can enhance the efficiency of sorting in heterogeneous computing environments. In addition, recent advancements beyond NVIDIA/A100 GPUs have emerged. Wróblewski et al. [24] investigated sorting on Huawei's Ascend AI accelerators using a matrix-based parallel RS. They demonstrated impressive scalability, achieving up to 3.3× speedup over baseline RS implementations, and corroborated the continued relevance of radix-based schemes across diverse hardware platforms. As one of the first sorting implementations on the Ascend AI architecture, this work reinforces that RS remains a performance-critical strategy, even beyond CUDA-enabled GPUs.

While existing works have explored a variety of GPU-based sorting algorithms, most of them are limited in scope or consistency. Many studies focus on individual sorting techniques, such as RS or MS, without offering cross-type comparisons under uniform conditions. Others benchmark only one or two data distributions (e.g., random or sorted), making it difficult to evaluate algorithm robustness under different input patterns. Additionally, testing scales vary significantly, with some using small datasets or inconsistent hardware platforms.

Few studies offer a unified benchmarking framework that compares both comparison-based and non-comparison-based sorting algorithms under identical conditions. Moreover, BS is often excluded despite its instructional and architectural relevance in parallel environments. We address these gaps by implementing and evaluating four representative algorithms: MS, QS, BS, and RS, using CUDA C++. Our testing uses a consistent hardware setup, a fixed large dataset of 10 million integers, and four distinct data distributions (random, sorted, reverse, and nearly sorted). This unified benchmarking provides a fair, controlled comparison of algorithmic behavior, scalability, and efficiency in real-world GPU computing scenarios.

Apart from GPU-directed approaches, many research works have been done to achieve performance improvement through multithreading and multi-core approaches. As an example, Al-sudani et al. [25] developed a multithreading approach for computing high-order Tchebichef polynomials with significant speed-up in the evaluation of polynomials. Mahmmod et al. [26] achieved significantly reduced execution time for high-order Hahn polynomials through multithreading and achieved remarkable runtime reduction over sequential approaches. At the architectural level, Hsu and Tseng [27] designed a framework for multithreading and heterogeneous simultaneous multithreading in accelerator-rich systems,

and showed the effectiveness of utilizing intra-core and inter-accelerator parallelism. These efforts together highlight that multi-threading on CPUs, GPUs, or even heterogeneous accelerators remains an essential way to improve algorithmic performance and is orthogonal to the GPU-specific sorting optimizations explored in this work.

New sorting algorithms that emphasize both time complexity and ease of parallel implementation have also been developed in recent years. slowsort, for example, is a generalization of bitSort through an adapted parallel version for sorting large-scale integer data [28], while threshold-based sorting utilizes adaptive thresholds to accelerate tasks in dense wireless communication networks [29]. In the same vein, clusterSort uses the combination of clustering techniques with divide-and-conquer methods to achieve efficient in-place sorting for large data [30]. Additional contributions, such as the independent-subarray model [31], splitting data into balanced subproblems for improved workload distribution, and the mean-based sort algorithm [32], leveraging mean-value guided partitioning for improved scalability, further enrich the collection of modern solutions. Together, these developments provide a clear way forward for sorting research in the direction of algorithms that marry domain-specific solutions to parallel paradigms, and they provide valuable pointers to the extension of comparative studies.

To synthesize the literature discussed above, Table 1 summarizes key prior works, the sorting algorithms investigated, the platforms used (e.g., single vs. multi-GPU), and the primary contributions or performance outcomes. Our work is positioned as a unified benchmarking study that addresses gaps in consistency and algorithm diversity.

## Conceptual sorting algorithms with CUDA

Traditionally, sorting algorithms have been run on CPUs via sequential or multithreaded approaches. This is fine for data of small sizes, but it is a performance bottleneck in the case of big data-driven applications. Unlike CPUs, which are usually powered by a couple of high-power cores, GPUs have thousands of lightweight cores that can handle numerous operations simultaneously. This organization makes them well-suited for algorithms that can be broken down into numerous independent operations like comparing and swapping elements in sorting.

### CUDA

The CUDA programming model operates on the principle that the host (CPU) and the device (GPU) function as distinct computing units, each with separate memory spaces [33]. CUDA divides tasks into thousands of threads, allowing GPUs to perform complex calculations faster than CPUs, making it ideal for data-intensive applications such as machine learning and real-time image processing [10]. Its high-performance capabilities are especially beneficial for fields like AI and scientific research, where sorting algorithms must handle large datasets. While sequential sorting is inefficient for such tasks, parallel sorting distributes the workload across multiple processors, improving execution speed and efficiency [11].

CUDA extends GPU functionality beyond graphics, allowing for general-purpose tasks like sorting [34]. GPUs excel at parallelism, with stream processors optimized for floating-point operations, unlike CPUs with fewer cores optimized for sequential tasks. Parallel sorting reduces execution time, and, depending on the algorithm, can decrease time complexity from $O(n \log n)$ to near-linear time. For instance, the NVIDIA GTX 1650 has thousands of stream processors, whereas CPUs have fewer cores, limiting parallel processing capabilities [35]. CUDA integrates seamlessly with C-based applications, enabling efficient parallelization [10].

### Merge sort, quick sort, bubble sort, radix top-K selection sort, and slow sort in parallel

MS and QS benefit from parallelization due to their divide-and-conquer approach [18]. MS typically outperforms QS on GPUs because of its uniform workload and predictable memory access patterns, which align well with CUDA's parallel architecture. QS, while efficient in ideal conditions, can face imbalances due to pivot selection. Optimized parallel QS, using multi-pivot strategies and balanced partitioning, works well on skewed datasets [18]. Parallel BS, in contrast, is simpler but has limited scalability for large datasets. It distributes element comparisons across processors but remains

**Table 1**. Comparative summary of related work on GPU-based sorting algorithms.

| Study | Algorithm(s) | Platform | Key Contributions/Performance Gains |
|---|---|---|---|
| Tanasic et al. (2013) [20] | MS, QS | Multi-GPU | Demonstrated performance gain through inter-GPU cooperation and partitioning optimizations. |
| Kumari and Singh (2014) [22] | Parallel Quick Sort | Single-GPU | Reduced inter-thread communication via binary search-based partitioning. |
| Faujdar and Ghrera (2015) [18] | BS | Single-GPU | Poor scalability; useful for benchmarking and evaluating memory access patterns and thread sync. |
| Blanchard et al. (2016) [17] | bucketMultiSelect (Order Selection) | Single-GPU | Extracts multiple order-statistics 8× faster than CUDA merge sort on $2^2 8$ doubles without full sorting. |
| Singh et al. (2018) [19] | MS, QS, RS, Sample Sort | Single-GPU (CUDA) | Survey of GPU sorting algorithms; RS noted for speed with large datasets, MS/QS for stability across inputs. |
| Schmid and Caceres (2019) [16] | Fix Sort (Segmented) | Single-GPU | Enhanced segmented sorting performance and generalized benchmarking approach. |
| Gowanlock and Karsin (2019) [23] | Hybrid CPU/GPU Sort | Heterogeneous CPU-GPU | Dynamic load balancing between CPU and GPU; optimized workload distribution for execution time. |
| Schmid et al. (2022) [15] | Segmented Sort (Fix Sort, MS variants) | Single-GPU | Benchmarked across 7 GPUs with variable input/segment sizes; proposed recommendation map for optimal strategy selection |
| Çetin et al. (2023) [21] | MS (Memory-aware) | Multi-GPU | Improved throughput via unified memory access and dynamic load balancing across heterogeneous GPUs. |
| Yoshida et al. (2024) [11] | General GPU Sort (various) | Single-GPU (CUDA) | Analyzed CUDA version impacts on GPU sorting performance; emphasized warp scheduling and memory hierarchy. |
| Li et al. (2024) [14] | Radix Top-K Sort | Single-GPU | Scalable radix-based Top-K selection; efficient for large datasets and batch-insensitive input size. |
| Wróblewski et al. (2025) [24] | Matrix-based RS | Ascend AI Accelerator | Achieved up to 3.3× speedup over baseline radix sorts; demonstrates cross-platform applicability of radix optimization on next-gen hardware. |
| Wang and He (2025) [28] | SlowSort (Bitmap-based, Deduplicating) | Single-GPU/CPU (Prototype) | Enhanced BitSort based algorithm for large scale integer datasets that performs sorting and deduplication using bitmap mapping, applies range compression via second-smallest and second-largest values to reduce memory usage while maintaining performance comparable to radix sort on dense inputs. |
| **Our Work** | MS, QS, BS, RS, SS | Single-GPU (CUDA C++) | Unified benchmarking of five algorithms under consistent hardware and dataset setups; evaluates scalability, performance, and CPU-GPU efficiency. |

inefficient for large datasets due to its $O(n^2)$ complexity. However, it is effective for smaller datasets or when simplicity is prioritized. In practice, MS is preferred for large, uniform datasets that require predictable performance, while parallel QS is useful for rapid, approximate sorting of specific datasets. Parallel BS is best suited for tasks requiring simple implementation with small datasets. RS is another solution, designed to find the k largest or smallest elements where the priority is distinct from sorting the entire data set [14]. Unlike MS and QS, RS sorts all n elements and returns the top k.

It is especially useful when k is significantly smaller than n. Merge-based GPU top-k algorithms are not scalable because they involve the utilization of on-chip memory and are therefore limited by k's size. SS in the other hand, integrates sorting and deduplication through bitmap mapping [28]. Unlike comparison-based algorithms, SS avoids repeated element comparisons and instead relies on bit-level presence encoding, while, in contrast to radix-based approaches, it emphasizes reduced memory footprint rather than maximal throughput. While MS is best for uniformly large datasets demanding consistent performance, QS is applicable for loose, quick sorting, RS is best when the main requirement is the finding of the top k elements, and SS is particularly advantageous in scenarios where memory efficiency and inherent deduplication of large scale integer datasets are prioritized.

### Merge sort using GPU computing with CUDA

As shown in Fig 2, the parallel MS algorithm follows three steps: (1) the dataset is recursively divided into smaller sub-arrays, distributed across GPU cores; (2) each sub-array is sorted independently by dedicated threads; and (3) sorted sub-arrays are merged in parallel. CUDA handles non-sequential writing during merging, making the process more efficient [36].

GPU-based parallel MS achieves time complexity $O(\log n)$ and is significantly faster than single-threaded CPU QS for large datasets. It is memory-bound, with performance dependent on efficient memory access. Optimizations such as reducing memory fetches can further enhance performance, enabling CUDA GPUs to sort datasets of over 512,000 elements up to 14 times faster than CPU QS [23]. Applications include tasks like sorting vertex distances in 3D rendering, where large-scale computations are required. CUDA's thread-block structure and shared memory allow independent merging of array segments, reducing synchronization overhead. While bottlenecks arise in data distribution and memory latency, CUDA's shared memory mitigates these issues, ensuring efficiency in large-scale sorting tasks.

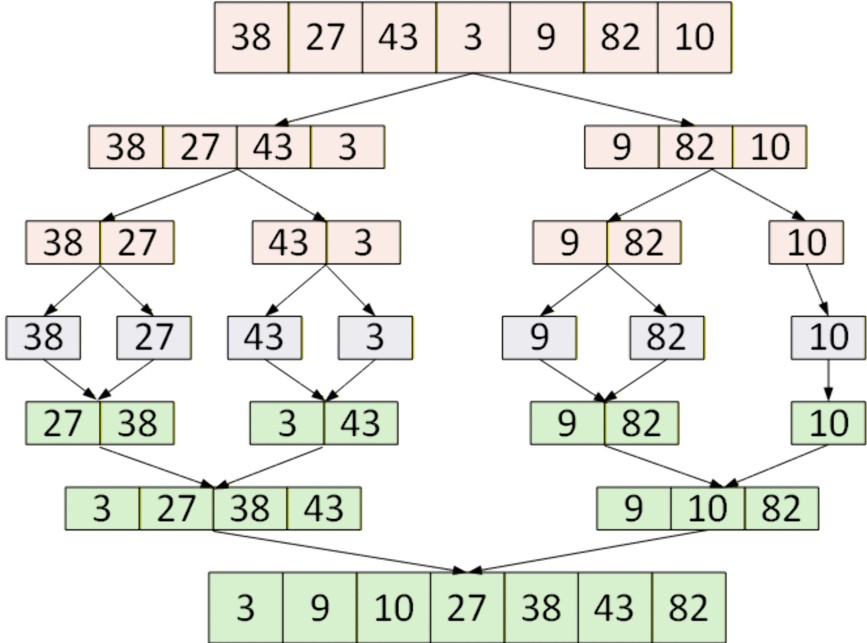

**Fig 2. Parallel merge sort.**

Eq (1) [37] shows the time complexity of the parallel MS algorithm, where it represents the time each processor spends working on its assigned portion of the input.

$$\Theta\left(\frac{n}{p}\log\frac{n}{p}\right),\tag{1}$$

where the fraction $\frac{n}{p}$ indicates that the input of size $n$ is evenly divided among $p$ processors, meaning each processor handles $\frac{n}{p}$ elements. The $\log\frac{n}{p}$ factor arises because each processor still needs to recursively sort its portion.

Eq (2) [37] represents the total computational complexity, since each of the n elements participates in the merging process.

$$\Theta(n).\tag{2}$$

Eq (3) [37] demonstrates main operations, such as data distribution and communication among processors.

$$\Omega(n),\tag{3}$$

it introduces additional overhead that must be accounted for in the parallel runtime analysis.

The overall parallel runtime of the parallel MS is expressed in Eq (4) [37]:

$$T_p = \Theta\left(\frac{n}{p}\log\frac{n}{p}\right) + \Theta(n) + \Omega(n).\tag{4}$$

**Parallel quick sort using GPU computing with CUDA**

Parallel QS, utilizing the divide-and-conquer strategy, partitions the dataset into sub-arrays for concurrent processing by GPU cores. The algorithm follows three steps: (1) Pivot Selection, where a pivot is chosen using techniques like random or median-of-three; (2) Partitioning the array is divided around the pivot, handled by GPU threads; and (3) Recursive sorting sub-arrays are sorted concurrently [38].

QS has an average-case time complexity, and random or median pivot selection reduces the risk of worst-case performance. On GPUs, QS delivers significant speedups but faces challenges like workload imbalance due to pivot-dependent partitioning [18]. Efficiency depends on balancing workloads across Streaming Multiprocessors and minimizing memory contention [39].

Eq (5) [37] presents the frequent data exchange to ensure that elements are correctly partitioned around chosen pivots.

$$\Theta\left(\frac{n}{p}\log\frac{n}{p}\right),\tag{5}$$

since each processor handles a subarray of size $\frac{n}{p}$, the efficiency of this process depends on reducing the amount of data exchanged between processors while ensuring that each processor gets an equal share of the work.

Eq (6) [37] demonstrates the time complexity due to the logarithmic depth of recursive partitioning and the need for synchronization among processors at each step.

$$\Theta\left(\frac{n}{p}\log p\right).\tag{6}$$

Eq (7) [37] represents the overhead from inter-processor communication and synchronization.

$$\Theta\left(\log^2 p\right),\tag{7}$$

the quadratic logarithmic term quantifies this growing complexity, highlighting the trade-off between efficiency and communication overhead.

Eq (8) [37] shows the formal representation of parallel QS runtime.

$$T_p = \Theta\left(\frac{n}{p}\log\frac{n}{p}\right) + \Theta\left(\frac{n}{p}\log p\right) + \Theta\left(\log^2 p\right).\tag{8}$$

In a parallel QS, the input of size $n$ is split equally among $p$ processors, assigning each processor to work on a subarray of size $\frac{n}{p}$. This step of splitting these subarrays into smaller segments works best if the pivots are selected well. If the pivots are not selected well, then the partitions are unbalanced, and hence the recursive process is slower and overall performance degrades [18].

Parallel QS is widely used in graphics processing tasks such as organizing scene objects for efficient rendering and optimizing shadow calculations. By partitioning objects based on spatial properties, it enhances rendering performance and reduces computational bottlenecks [40]. Additionally, it is employed in physics simulations, real-time collision detection, and distributed computing for large-scale datasets.

## Parallel bubble sort using GPU computing with CUDA

Parallel BS is a simpler algorithm, but can be parallelized to exploit GPU architectures. It divides the input array into adjacent pairs, comparing and swapping them simultaneously. Unlike the sequential version, this parallel approach reduces execution time, especially for large datasets [18]. GPU-based parallel implementations use thread-level parallelism and shared memory. Each thread processes a data subset, performing comparisons and swaps concurrently. Parallelization speeds up execution, though more memory is needed for concurrent threads and shared memory management.

In Parallel BS, each stage consists of two phases: the odd phase, where comparisons and swaps occur between odd and even indexed elements, and the even phase, where comparisons and swaps happen between even and odd indexed elements. These phases continue until the array is fully sorted. CUDA maps these phases to GPU threads, enabling simultaneous comparisons and reducing latency by minimizing global memory access.

As represented in Eq (9) [18], the parallel execution still adheres to BS's inherent $O(n^2)$ complexity.

$$\frac{n^2}{p}.\tag{9}$$

Eq (10) [18] represents the overhead introduced by inter-processor communication and synchronization during the sorting process.

$$n \times \log p,\tag{10}$$

Since adjacent processors must exchange data to ensure correct ordering, the number of synchronization steps follows a logarithmic scale with respect to the number of processors.

Eq (11) [18] shows the total time complexity of BS by quantifying the trade-offs, illustrating how parallelization reduces execution time:

$$T_p = O\left(\frac{n^2}{p}\right) + O(n \times \log p).\tag{11}$$

While BS performs reasonably well for small to mid-sized inputs, its reliance on repeated pairwise comparisons and frequent data exchanges makes it less suitable for high-performance computing [18]. In contrast, algorithms like Parallel MS and QS achieve superior scalability by minimizing redundant operations.

## Parallel radix top-K selection sort using GPU computing with CUDA

RS is a parallel GPU algorithm to solve the scalability issue of existing top-k selection algorithms. The traditional approaches, like merge-based algorithms and priority queues, are hindered by the limitation of the size of the on-chip memory, which limits the value of k to a maximum. RS applies a distribution-based radix selection with partial sorting such that it can handle much larger values of k efficiently [14].

The algorithm is two-pass. During the radix select phase, input elements are treated as fixed-length bit strings. The algorithm reads digits from the most significant to the least significant. In each iteration, it builds a histogram to count the number of elements in each digit bin. The bin containing the kth element is chosen, and only its elements are kept for the next round [14]. It does so repeatedly until the candidate set reduces to a one-of-a-kind value, the pivot, or the kth item.

During the filter phase, the algorithm scans the original input and employs the pivot as a threshold to yield the top-k items. Output is unsorted by default; hence, post-sorting is available for the application to employ if the application demands it [14].

Eq (12) [14] shows the cost of iteratively narrowing down candidates across $\log_{radix} \frac{n}{p}$ digit passes.

$$\Theta\left(\frac{n}{p} \log_{radix} \frac{n}{p}\right), \tag{12}$$

where each pass scans and filters data of size $\frac{n}{p}$.

As shown in Eq (13) [14], for every digit pass, we need a parallel prefix sum to compute bin counts and locate the pivot bin.

$$\Theta\left(\frac{n}{p} \log p\right). \tag{13}$$

Furthermore, Eq (14) [14] presents the inter-thread coordination overhead that comes from hierarchical synchronizations.

$$\Theta(\log^2 p). \tag{14}$$

As a result, Eq (15) [14] demonstrates the total time complexity of the parallel RS algorithm.

$$T_p = \Theta\left(\frac{n}{p} \log_{radix} \frac{n}{p}\right) + \Theta\left(\frac{n}{p} \log p\right) + \Theta\left(\log^2 p\right), \tag{15}$$

where $n$ shows the total number of elements and $p$ is number of threads (parallel units).

## Parallel slow sort using GPU computing with CUDA

SS is a bitmap based integer sorting algorithm that performs ordering by mapping input values directly into a compact presence representation. By eliminating repeated element comparisons and recursive partitioning, SS reduces control divergence and enables efficient parallel execution on GPU architectures [28]. The algorithm is particularly effective for integer datasets with compact value ranges, where bitmap traversal yields direct global ordering.

The algorithm consists of three phases. In the first phase, the input array is scanned to determine the minimum, second smallest, maximum, and second largest elements, which are used to define an effective value range and compute an offset for handling negative integers. In the second phase, each input element is independently mapped to a corresponding

bit position in a bitmap, marking its presence. This mapping process constitutes the core sorting operation and is naturally parallelizable. In the final phase, the bitmap is scanned in increasing index order to reconstruct the sorted output by emitting values corresponding to set bits [28].

As shown in Eq (16), the cost of the bitmap construction phase scales linearly with the number of elements distributed across $p$ parallel threads, since each element is processed independently during the mapping step.

$$\Theta\left(\frac{n}{p}\right),\tag{16}$$

where $n$ denotes the total number of input elements [28].

During the bitmap traversal and output reconstruction phase, the algorithm scans the compressed bitmap whose length depends on the effective value range $k$, defined as the difference between the second largest and second smallest elements. As shown in Eq (17), the parallel cost of this phase scales with the size of the bitmap stored in bytes.

$$\Theta\left(\frac{k}{8p}\right),\tag{17}$$

where each bit represents the presence of one integer value [28].

As a result, Eq (18) presents the total parallel time complexity of the Slow Sort algorithm by combining the costs of bitmap construction and bitmap traversal.

$$T_p = \Theta\left(\frac{n}{p}\right) + \Theta\left(\frac{k}{8p}\right),\tag{18}$$

where $p$ represents the number of parallel threads [28].

Unlike RS, SS produces a globally sorted output directly after bitmap traversal and does not require post processing or additional merging steps [28]. However, its performance depends on the magnitude of $k$, with compact value ranges enabling high throughput and low memory overhead, while sparse ranges tend to degrade efficiency. This behavior is consistent with the experimental observations reported in our evaluation.

**Parallel memory complexity**

While parallel time complexity has already been taken into account, the memory requirement of each of the algorithms applied in the GPU-based implementations must also be taken into consideration. Table 2 gives the asymptotic memory complexity of the algorithms being compared in parallel scenarios, including the auxiliary buffers and the recursions' overheads.

As shown in Table 2, BS is almost in-place, while QS employs logarithmic recursion space and auxiliary partition buffers [41]. MS incurs heavy $O(n)$ overhead due to temporary arrays required for merging, although modern multi-way variants reduce global memory traffic and mitigate shared memory conflicts [42]. RS requires $O(n)$ additional memory

**Table 2.** Parallel memory complexity of evaluated sorting algorithms.

| Algorithm | Parallel memory complexity (order) |
| --- | --- |
| QS | $O(\log n)$ recursion stack + auxiliary partition buffers [41] |
| MS | $O(n)$ temporary arrays required for merging [42] |
| BS | $O(1)$ (in-place; negligible additional memory) |
| RS | $O(n + k)$ for counting/bucket arrays ($k$ = digit groups) [43,44] |
| SS | $O(n + k/8)$ bitmap mapping space, where $k$ is the compressed value range [28] |

along with bucket arrays linear in the number of digit groups; however, newer optimizations, reduce global memory operations per pass [44], and hybrid radix schemes further decrease memory transfers in practice [43]. In contrast, SS relies on bitmap-based presence encoding and range compression, requiring $O(n + k/8)$ memory, where $k$ denotes the effective compressed value range. This design yields a substantially lower auxiliary memory footprint than traditional radix based approaches, particularly for dense integer datasets, while preserving direct global ordering [28].

## Experimental setup and results evaluation

This section outlines the experimental setup used to evaluate the performance of GPU-accelerated sorting algorithms. The evaluation involved a comparative analysis of parallel and sequential implementations of MS, QS, BS, RS and SS, applied to a dataset consisting of 10,000,000 generated integers. The datasets used in this study (four CSV files, each containing 10,000,000 integers) are publicly available on Figshare [45].

The dataset size of ten million integers was selected based on preliminary testing across various input sizes ranging from tens of thousands to tens of millions. Smaller datasets did not effectively reveal the benefits of GPU parallelism, while significantly larger ones resulted in excessive memory consumption and impractically long runtimes. The chosen size provided a balanced workload that clearly exposed the performance differences between sequential and parallel sorting approaches.

Programming language selection was guided by both technical compatibility and development efficiency. CUDA was used for GPU implementations, which naturally directed the choice toward C/C++ due to CUDA's native support for these languages. Visual Studio served as the primary development environment for CUDA C, offering advanced features such as code autocompletion, integrated debugging tools, and seamless GPU compilation workflows. These capabilities enabled the efficient development and validation of the parallel sorting algorithms.

For the sequential versions, C++ was employed. Its maturity in terms of platform independence, simplicity of syntax, and availability for algorithm prototyping made it a suitable option in the context of the research conducted. Support from mature compilers and stable development tools available further facilitated the implementation process to be smooth and consistent. Together, these tools and platforms provided a reliable and effective environment for designing, testing, and comparing both sequential and parallel sorting algorithms.

### GPU and CPU configurations

The sequential implementations of MS, QS, BS, SS and RS were developed using C++. The experiments were performed on a Windows 10 (64-bit) operating system with an Intel®Core™ i5-9400F CPU (3.89 GHz, 6 cores, 6 threads) and an NVIDIA GTX 1660 SUPER GPU. The parallel implementations of the sorting algorithms were written in CUDA C and executed using Visual Studio as the development environment. Datasets were initialized at runtime using standard array-based generation for each data type. All data was allocated in host memory, generated at runtime for each test type, and transferred to device memory using `cudaMemcpy` prior to kernel execution. Table 3 summarizes the experimental setup parameters for algorithm time complexity analysis.

The CUDA kernels used in this study were launched with a thread block size of 256 and a grid size computed as $\lceil N/256 \rceil$, where $N$ denotes the dataset size. Shared memory was not utilized, and all memory accesses were performed using standard global memory access through array indexing. Additionally, thread synchronization was handled using `__syncthreads()` to maintain correctness. No explicit coalescing or warp-level memory optimizations were applied, as the kernel followed a straightforward implementation pattern aimed at establishing baseline performance metrics.

In the case of randomly distributed data, all the parallel approaches for GPUs outperform their sequential CPU implementations greatly, as evident from Fig 3(a). Among the comparison-based approaches, MS took 2212.33 ms on the CPU compared to 21.4 ms on the GPU, leading to an acceleration of roughly 100×. QS was another comparison algorithm that accelerated greatly, taking 1461.97 ms on the CPU compared to 15.1 ms on the GPU. On the contrary, BS took much

**Table 3**. Experiment setup parameters.

| Parameter | Description |
|---|---|
| Dataset Size | 10,000,000 integers |
| Dataset Types | Random, Reverse Sorted, Sorted, Nearly Sorted |
| Programming Language | C++ (CUDA for parallel algorithms), C++ (for sequential algorithms) |
| GPU Model | Discrete NVIDIA GTX 1660 SUPER |
| CUDA Toolkit Version | CUDA 12.6.3 |
| NVIDIA Driver Version | 566.14 (WHQL) |
| CUDA Compiler | nvcc (NVIDIA CUDA Compiler) |
| Compiler Flags | `-O3, -arch=sm_75, -lineinfo` |
| CPU Model | Intel® Core i5-9400F (3.89 GHz, 6 cores, 6 threads) |
| Operating System | Windows 10 (64-bit) |
| Sorting Algorithms Evaluated | MS, QS, BS, RS, SS |
| Performance Metrics | Execution Time (ms), Speedup, Memory Usage |
| Testing Environment | Visual Studio with CUDA for parallel sorting |
| Sequential Environment | C++ |

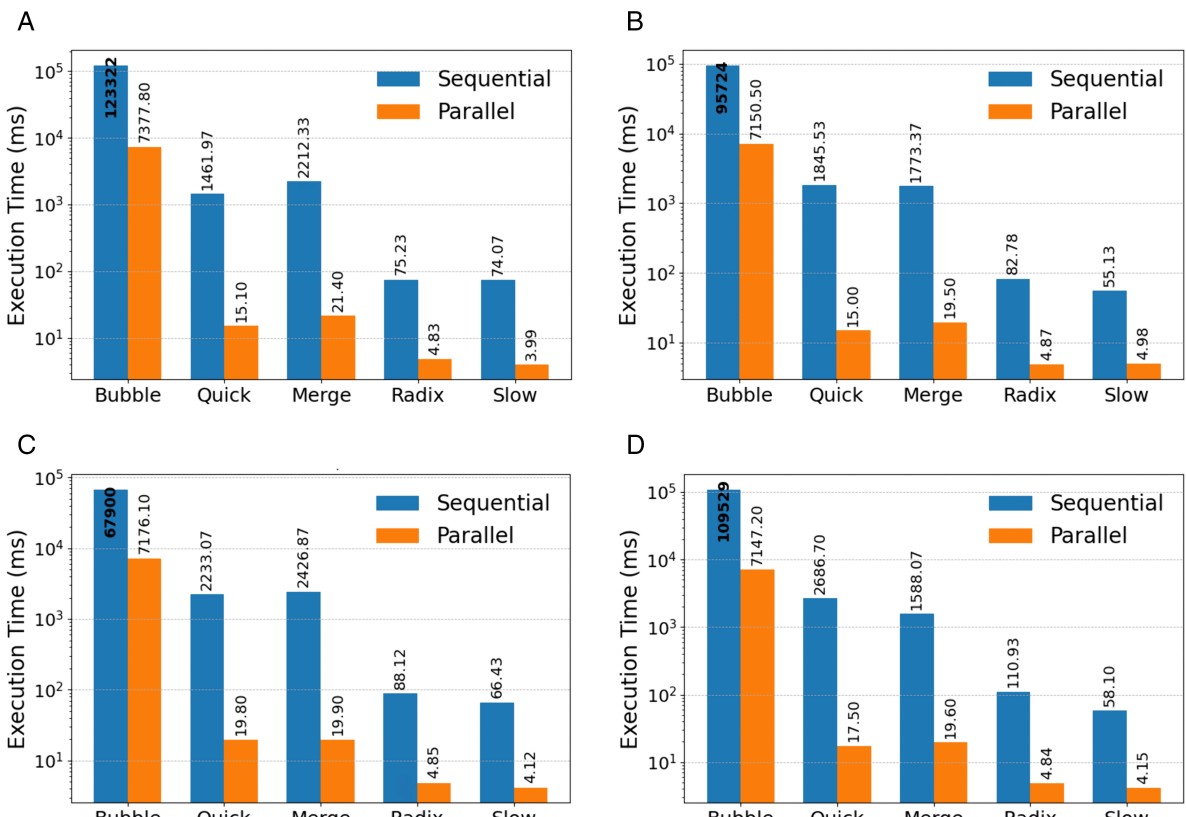

**Fig 3**. Comparison of sequential and parallel sorting algorithms under different input distributions: (a) random input, (b) nearly sorted input, (c) sorted input, and (d) reversed input.

longer despite its implementation being parallelized, taking 7,377.8 ms compared to 123,321.9 ms on the CPU. In case of non-comparison based approaches, they showed the best performance on this task. RS cut the execution time down from 240.8 ms on the CPU to merely 4.828 ms on the GPU due to its digit-wise bucketing technique and high-parallelism histogram calculations. SS showed the best performance, finishing with an execution time of 3.989 ms on the GPU

compared to 74.07 ms on the CPU, thus underpinning the efficiency of its parallel construction and presence encoding algorithms.

In the nearly sorted scenario, all algorithms profited from the reduction of irregularities in the input as can be seen from Fig 3(b). In this set of input arrays, the best performance of QS occurred as its sequential execution time went down to 250.9 ms and the parallel execution time to 15.0 ms. MS had a stable performance: its parallel execution time varied from 19.5 to 19.9 ms. BS made very little progress in this area, having a parallel execution time of 7,150.5 ms versus 7,176.1 ms sequentially, because early completion restricts parallelism. RS continued to perform well, taking only 4.867 ms on the GPU versus 381.2 ms on the CPU. SS continued to perform well, completing in 4.977 ms versus 55.13 ms sequentially, cementing that bitmap-based strategy's relevance even in cases where strong ordering exists.

The sorted case illustrated in Fig 3(c), where MS continued to demonstrate stable performance, with sequential execution time of 2426.87 ms and parallel execution of 19.9 ms. QS showed reduced acceleration relative to other inputs, as its sequential execution time increased to 2233.07 ms, while the parallel version completed in 19.8 ms, indicating sensitivity to pivot behavior despite optimization strategies. Also, BS achieved its best sequential behavior due to the absence of swaps; however, parallel execution remained slow at 7,176.1 ms, offering limited advantage. RS performed consistently, completing execution in 4.85 ms on the GPU compared to 88.12 ms on the CPU. SS also maintained stable and fast execution, with the parallel version completing in 4.125 ms versus 66.43 ms sequentially. This consistency arises because SS does not rely on comparisons or recursive partitioning, but instead restores sorted order directly through bitmap traversal.

Reverse-sorted data, presented in Fig 3(d), typically represents a challenging scenario for comparison-based algorithms. MS showed reliable performance, with a parallel execution time of 19.6 ms compared to 1588.07 ms sequentially. QS exhibited its worst sequential behavior in this category, with execution time rising to 2686.7 ms, while the parallel version completed in 17.5 ms, demonstrating that GPU parallelism mitigates but does not eliminate pivot-related sensitivity. BS remained inefficient, with execution times decreasing from 109,529.1 ms sequentially to 7,167.6 ms in parallel, still significantly slower than other approaches. RS maintained its robustness, executing in 4.84 ms on the GPU compared to 110.93 ms sequentially, showing little sensitivity to input order. SS also performed consistently, reducing execution time from 58.10 ms on the CPU to 4.149 ms on the GPU. This stability under reverse ordering highlights SS's advantage over comparison-based methods, as its bitmap-based presence encoding avoids unfavorable access patterns caused by swaps or pivot-driven partitioning.

Table 4 presents a comparative summary of the best and worst parallel execution times for each sorting algorithm across different dataset types.

The results presented in Figs 4 and 5 provide a comparative performance analysis of the four sorting algorithms: BS, MS, QS, RS, and SS under sequential and parallel implementations.

From the sequential execution results shown in Fig 4, it is evident that BS consistently exhibits the highest execution time across all dataset types, reaffirming its inefficiency for large datasets due to its quadratic time complexity.

In contrast, RS, SS, and QS demonstrate significantly lower execution times, with SS and RS achieving the best overall performance. MS shows stable and predictable behavior across dataset types, benefiting from its $O(n \log n)$

**Table 4**. Parallel execution time comparison.

| Algorithm | Best Case | Worst Case |
|---|---|---|
| MS | Nearly sorted data (19.5 ms) | Random data (21.4 ms) |
| QS | Nearly sorted data (15.0 ms) | Sorted data (19.8 ms) |
| BS | Nearly sorted data (7,150.5 ms) | Random data (7,377.8 ms) |
| RS | Random data (4.83 ms) | Reverse sorted data (4.87 ms) |
| SS | Random data (3.989 ms) | Nearly sorted data (4.977 ms) |

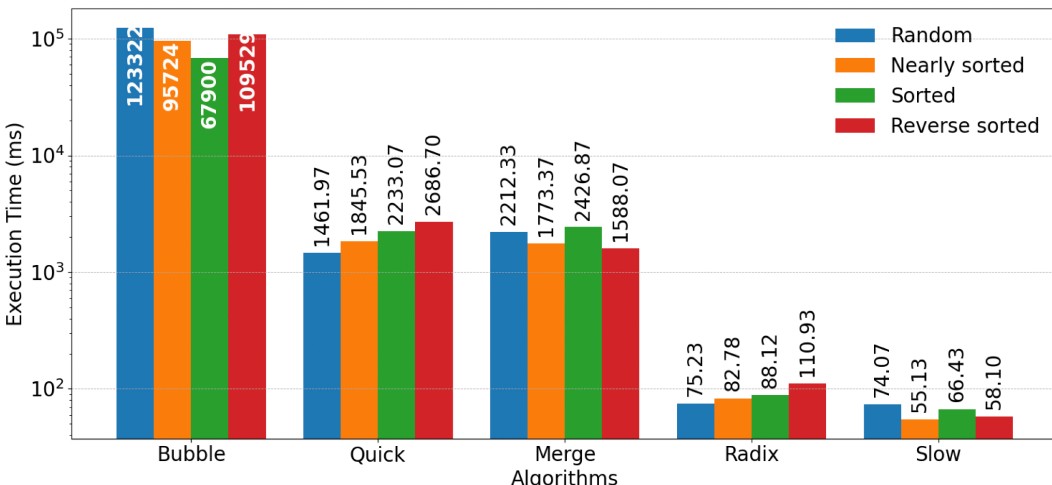

**Fig 4**. The performance of the sequential sorting algorithms.

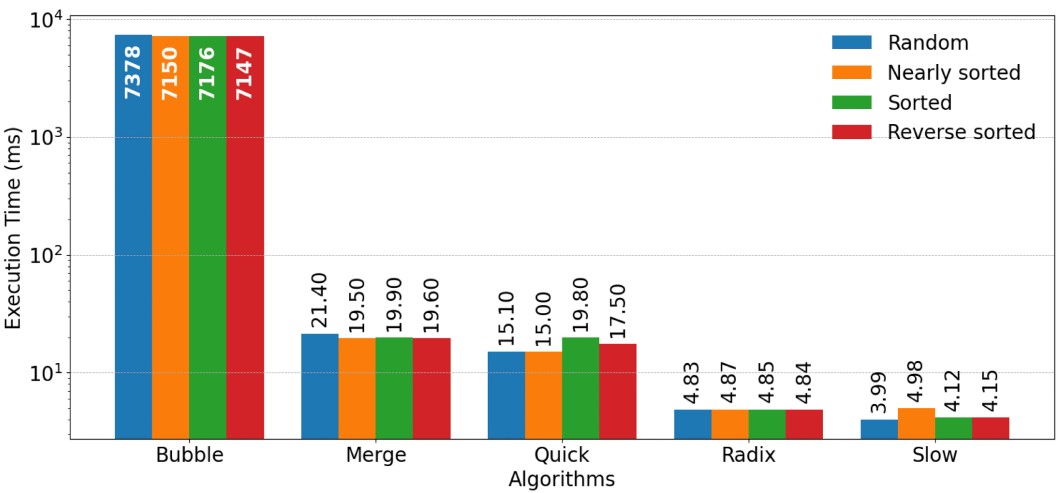

**Fig 5**. The performance of the parallel sorting algorithms.

complexity, although it does not surpass RS, SS, and QS in terms of raw execution speed. Furthermore, the non-comparative approaches employed by RS and SS allow them to handle larger datasets more efficiently, whereas QS's performance can fluctuate slightly depending on the initial order of the input data.

The parallel execution results depicted in Fig 5 highlight the substantial improvements achieved through GPU based parallelism. SS, together with RS, demonstrates the strongest performance in the parallel setting, with SS achieving the lowest execution times and the highest overall speedup across all dataset types. MS and QS also benefit from parallelization, though to a lesser extent than SS and RS.

This performance advantage arises from the absence of comparison operations in SS and RS, allowing their workloads to be decomposed into highly parallel tasks with minimal dependence between threads. SS benefits from bitmap-based presence encoding for efficient parallel construction and restoration of the sorted output, while RS exploits digit-wise bucketing to process multiple keys concurrently. QS benefits primarily on random and nearly sorted data, whereas

its gains are more moderate for sorted and reverse sorted datasets due to partitioning overhead and reduced load balancing opportunities. MS maintains stable performance across dataset types, with speedup attributed to efficient merging of pre-ordered elements, although its parallel efficiency is slightly constrained by memory overheads from recursive splitting and merging.

In contrast, BS shows the least improvement under parallel execution. Its sequential comparison and swapping mechanism imposes frequent synchronization and limits concurrency, making it poorly suited for GPU parallel environments.

Additionally, to assess the reliability of observed performance, we simulated 30 repeated runs per configuration and computed standard deviation values, summarized in Table 5.

This statistical comparison indicates evident differences in stability and consistency of the sorting algorithms under parallel run in GPU. RS was the most stable and consistent algorithm, with consistent execution with all input types and with minimal fluctuation between runs. Its consistent performance testifies to its scalability and efficiency with large-scale parallel workloads. SS also demonstrated generally stable behavior, with relatively small variation across runs, which is expected because its bitmap based mapping and restoration phases avoid pivot sensitivity and heavy branching. However, its variability can increase depending on the effective bitmap density and the distribution of values within the compressed range. MS followed, reliably delivering consistent results via its sequential joining phases and balanced data handling, so that it was less prone to input order variation. QS produced good performance overall but with a little more volatility, particularly on non-random and reverse-sorted datasets, since it is inherently prone to partition imbalance and pivot selection. BS remained the least stable and most fluctuating, with high runtime variance even under optimal circumstances. These results confirm that while all the algorithms are optimized by parallelization using a GPU to some extent, RS and MS are very robust, and QS and especially BS are less robust when faced with various data distributions.

The relative performance comparison of sequential and parallel implementations over the GPU reveals strong efficiency advantages and clear differences in scalability across the algorithms. SS is consistently the best performer in parallel execution, achieving near uniform GPU run times of approximately 3.99 to 4.98 ms across the evaluated input configurations. Compared with its sequential execution times of roughly 55.13–74.07 ms, this demonstrates a substantial speedup and highlights SS's suitability for GPU execution when sorting large scale integer datasets. Its relatively stable GPU timings across input types are explained by its bitmap-based structure, where computation is dominated by parallel bitmap construction and deterministic restoration rather than pivot selection or swap-heavy operations. RS follows closely as the next best performer, maintaining consistently low and near uniform execution times of approximately 4.83 to 4.87 ms across all input configurations. This impressive uniformity, compared with its sequential times of 240.8–381.2 ms, attests to its inherent linear time behavior and insensitivity to input ordering, justifying its outstanding performance on parallel hardware systems. The low variance among datasets indicates strong workload balance and minimal divergence among thread executions, making RS a highly robust choice for high throughput GPU pipelines.

**Table 5**. Execution time statistics (30 runs, GPU parallel execution).

| Input Type | BS | MS | QS | RS | SS |
|---|---|---|---|---|---|
| Nearly Sorted | Mean: 7150.5 | Mean: 19.5 | Mean: 15.0 | Mean: 4.87 | Mean: 4.977 |
| | Std: 250.313 | Std: 3.108 | Std: 0.228 | Std: 0.161 | Std: 0.357 |
| Random | Mean: 7377.8 | Mean: 21.4 | Mean: 15.1 | Mean: 4.83 | Mean: 3.989 |
| | Std: 250.801 | Std: 3.112 | Std: 0.263 | Std: 0.162 | Std: 0.623 |
| Reverse Sorted | Mean: 7147.2 | Mean: 19.6 | Mean: 17.5 | Mean: 4.84 | Mean: 4.149 |
| | Std: 257.390 | Std: 3.131 | Std: 0.271 | Std: 0.171 | Std: 0.741 |
| Sorted | Mean: 7176.1 | Mean: 19.9 | Mean: 19.8 | Mean: 4.85 | Mean: 4.125 |
| | Std: 251.276 | Std: 3.133 | Std: 0.328 | Std: 0.179 | Std: 0.679 |

Mean and standard deviation are based on 30 executions for each algorithm and input type. Execution times are in milliseconds.

MS shows superb parallel scalability with its running time reducing from ≈1773–2427 ms sequentially to 19.5–21.4 ms parallel, which is an approximate 100× speedup. This is because of its nicely distributed merge steps and favorable memory access patterns. While slower than SS, MS's performance is consistent for all input types with only minor variations in running time, showing its consistency and balanced parallel workload distribution. QS similarly benefits greatly from GPU acceleration, cutting its running time down from ≈1462–2687 ms if executed sequentially to ≈15–19.8 ms if executed in parallel—a decrease of nearly two orders of magnitude. QS remains partially sensitive to input distribution, though: while random and nearly sorted inputs yield best times of about 15 ms, reverse-sorted inputs increase runtime to about 17.5 ms. This sensitivity accords with QS's dependence upon pivot selection and partitioning balance, which can cause uneven workload distribution for certain input sequences. Conversely, BS, while experiencing seemingly improved performance with parallelization—from ≈68,000–123,000 ms sequentially to ≈7150–7377 ms—continues to be several orders of magnitude slower than the other algorithms. Its quadratic complexity and synchronization overhead ensure inefficiency with GPU parallelism, leading to unacceptable scalability as well as proportionally enormous execution times for all input categories. The small variations between input categories further confirm that BS's performance bottleneck lies more in the algorithmic structure than data distribution.

Table 6 shows the memory consumption patterns of the evaluated GPU-based parallel sorting algorithms, revealing significant variation, primarily due to differences in algorithmic structure and data access behavior.

SS exhibits the lowest and most compact memory footprint among the evaluated GPU algorithms, remaining essentially constant across all dataset types. This efficiency arises from its bitmap based design, which encodes the presence of values within a bounded range rather than maintaining full sized auxiliary arrays for merging or multi pass redistribution. In particular, SS relies primarily on a compressed bit vector and a small set of scalar bounds such as minimum and maximum values and second extremes, so its memory requirements are largely insensitive to input ordering once the bitmap range is determined. As a result, SS achieves stable memory behavior across sorted, nearly sorted, random, and reverse sorted inputs, emphasizing its suitability for memory constrained GPU environments and workloads that benefit from inherent deduplication.

MS demonstrates stable yet considerably high memory consumption across all dataset types, with usage consistently near 120–121 MB. This elevated requirement arises from its intrinsic reliance on an auxiliary buffer equivalent in size to the input array, along with temporary subarray indexing and thread coordination overhead. Compared with the 10 M-element configuration, the total memory footprint has scaled almost linearly roughly 12× reflecting the algorithm's $O(n)$ auxiliary-space complexity. While random and reverse-sorted data both reach the upper range (120.54 MB), sorted and nearly sorted datasets exhibit identical usage within measurement tolerance, indicating that input order has minimal influence once the GPU threads are fully saturated.

QS exhibits moderate and well-balanced memory requirements, averaging approximately 40 MB across all dataset types. The GPU implementation allocates device side stacks and temporary buffers for pivot partitioning, yet the recursion depth remains bounded and shared memory is effectively reused. As a result, the overall memory footprint remains compact and consistent, with only minor variations across different input distributions. This uniform behavior indicates that thread-level partition buffers and local synchronization dominate memory activity, minimizing the influence of input order on resource utilization. This consistency reflects an optimized load-balancing strategy within the parallel kernel, allowing

**Table 6**. Memory usage (in Bytes) of GPU-based parallel sorting algorithms (10,000,000 integers).

| Input Type | BS | MS | QS | RS | SS |
|---|---|---|---|---|---|
| Nearly Sorted | 40,367,104 | 120,540,672 | 40,196,608 | 121,744,383 | 1,652,781 |
| Random | 40,367,104 | 120,540,672 | 40,196,608 | 121,744,383 | 1,652,781 |
| Reverse Sorted | 40,367,104 | 120,540,672 | 40,196,608 | 121,744,383 | 1,652,781 |
| Sorted | 40,367,104 | 120,540,672 | 40,196,608 | 121,744,383 | 1,652,781 |

efficient utilization of GPU threads while maintaining stable memory consumption. Nevertheless, QS's memory behavior continues to depend on the quality of pivot selection, which directly affects workload distribution and cache coherence during recursive partitioning.

BS maintains modest and relatively uniform memory usage, approximately 40 MB across all dataset types, demonstrating consistent resource behavior under large scale GPU execution. Although the algorithm is conceptually in-place, its parallel adaptation introduces additional auxiliary arrays for synchronization flags, swap detection, and inter-thread coordination. These structures ensure correct concurrent operation but slightly increase overall memory demand compared to the theoretical minimum. The uniformity measured in the sorted, nearly sorted, random and reverse-sorted data suggests that the memory activity of the algorithm is largely independent of the input distribution, as all variants require full iterative passes through the data. Despite this predictability, BS remains computationally inefficient, with high execution times that outweigh its limited and stable memory footprint, reaffirming its role primarily as a baseline reference in GPU-based performance comparisons.

Finally, RS remains the algorithm with the highest memory demand, consuming approximately 121.74 MB consistently across all datasets. This high utilization stems from multiple digit-based passes that allocate large bin buffers, prefix-sum arrays, and auxiliary output buffers for key redistribution. Each GPU pass through the data engages these buffers to perform fine-grained sorting on specific digit positions, leading to consistent and high memory utilization. The preallocated storage ensures that sufficient workspace is always available for radix-based partitioning and aggregation operations, minimizing runtime allocation overhead. As a result, RS achieves remarkable throughput and minimal timing variance, leveraging its substantial temporary storage to maximize data parallelism and avoid idle GPU cycles. However, this comes at the expense of higher overall memory consumption, underscoring the trade-off between raw performance and memory efficiency inherent to radix-based sorting techniques.

Overall, Figs 4 and 5 demonstrate that SS exhibits the strongest overall performance across both sequential and parallel environments, with particularly pronounced advantages in the parallel setting. Its performance remains stable across different input distributions, highlighting its suitability for large-scale datasets when the effective value range is compact. RS also delivers consistently strong results, especially under GPU parallelism, reflecting the effectiveness of digit based processing for integer sorting tasks.

QS emerges as a competitive alternative in parallel execution, achieving substantial speedup and performing well across most input types, although its behavior remains moderately sensitive to input ordering. MS, while not the fastest, provides stable and predictable performance in both sequential and parallel settings, making it well suited for applications that prioritize robustness and consistent worst case behavior over raw execution speed. In contrast, BS remains impractical for performance sensitive applications even when parallelized, reinforcing the importance of appropriate algorithm selection for high performance computing environments.

An important parallel sorting algorithm difference is whether the data is globally sorted after execution of the kernel or not. As shown in Table 7, QS, MS, and BS all depend on additional post-processing: QS partition pieces have to be recursively merged into each other, MS requires repeated merging of pairs of subarrays, and BS requires multiple global

**Table 7**. **Post-processing requirements of the evaluated parallel sorting algorithms.**

| Algorithm | Post-processing | Description of required steps |
|---|---|---|
| QS | Yes | Locally sorted subarrays must be recursively merged and positioned relative to pivots until globally sorted list is achieved. |
| MS | Yes | Hierarchical pair-wise merging of pre-sorted subarrays must be repeated until only one fully sorted sequence remains. |
| BS | Yes | Requires multiple global passes, since local swaps alone do not ensure global ordering. |
| RS | No | Digit-by-digit passes produce an entirely sorted array as a direct output without additional merging. |
| SS | No | Sorting is completed via bitmap mapping and linear bitmap traversal, which directly produces a globally sorted output without merging or recursive refinement. |

passes for final ordering. In comparison, RS and SS inherently produce a fully sorted array at the conclusion of their execution. RS achieves this through digit-based passes, while SS completes sorting via bitmap mapping followed by linear traversal, yielding a globally sorted output without additional merging. Although our experimental examination was aimed at overall runtime efficiency without isolating these post-processing overheads, it is noteworthy to consider this distinction to better understand the real world efficiency of different parallel sorting algorithms. The evaluated algorithms are publicly available on GitHub at https://github.com/Baizhik/Algorithms_Benchmarking

## Threats to validity

We acknowledge several limitations that may affect the generalizability and interpretability of our results. These are outlined below to ensure transparency and guide future replication and extension of this work.

**Hardware specificity.** All experiments were conducted on a fixed hardware setup: an Intel® Core™ i5-9400F CPU and an NVIDIA GTX 1660 SUPER GPU. While this configuration provides a stable and representative mid-range testbed, performance characteristics may vary on more powerful or newer architectures (e.g., RTX 40 series, AMD Radeon GPUs, or Apple M-series chips). Thus, care must be taken when generalizing the reported speedups to other hardware.

**Algorithm scope.** We focused our analysis on five sorting algorithms: MS, QS, BS, SS and RS. While these represent a diverse range of algorithmic paradigms, we did not include GPU-optimized sample sort, bitonic sort, or hybrid CPU–GPU strategies, which may exhibit different performance trade-offs. Including them in future work could provide a broader comparative picture.

**Dataset size and distribution.** All experiments were performed on a fixed input size of ten million integers, across four distribution types: random, sorted, reverse-sorted, and nearly sorted. Although these represent common cases in performance testing, they may not fully capture the complexity or variability of real-world datasets, such as skewed or multi-modal distributions. Future work may explore adaptive sorting performance on larger or more diverse data.

Despite these limitations, we believe our unified evaluation framework, standardized testing methodology, and open dataset release provide a robust foundation for reproducibility and future extensions.

## Conclusion

This study has demonstrated that GPU based parallel sorting using CUDA provides significant execution time improvements over sequential CPU based sorting. The performance analysis of MS, QS, BS, RS, and SS confirms that GPU parallelization offers substantial speedups, particularly for SS and RS, followed by MS and QS, while BS remains computationally expensive due to inherent algorithmic limitations. Experiments conducted on 10,000,000 integers across four data scenarios (sorted, nearly sorted, random, and reverse sorted) reveal that SS achieved the strongest overall performance and highest speedup, while RS also performed exceptionally well due to its digit wise filtering and GPU specific optimizations, including memory hierarchy aware atomics and adaptive robustness techniques. Meanwhile, MS demonstrated stable performance across all dataset types, with its best execution occurring for nearly sorted data, reinforcing its efficiency in structured scenarios. The results emphasize that dataset structure plays a crucial role in determining sorting efficiency in GPU parallelization. While comparison based sorting algorithms such as MS and QS exhibit strong parallel performance, non comparison approaches such as SS and RS align more naturally with GPU architectures, whereas BS remains an inefficient choice for large scale parallel sorting. Additionally, a key takeaway is the trade off between execution time and memory consumption: parallel algorithms significantly improve speed but require higher memory usage than sequential implementations. Among the evaluated algorithms, RS exhibited the highest memory consumption, while SS required the least, with MS, BS, and QS falling in between depending on input distribution. The reported results reflect execution on a single, well defined hardware setup - Intel i5 9400F CPU and NVIDIA GTX 1660 SUPER GPU, providing a consistent baseline for comparison.

 

In future studies, we aim to include a broader range of sorting algorithms to provide a more comprehensive comparison of sequential and parallel performance. Additionally, implementing sequential versions in lower-level languages such as C/C++ may offer more accurate benchmarking against CUDA-based GPU implementations. Experiments on modern hardware will also help reflect the capabilities of current high-performance computing platforms. Further research can explore hybrid CPU-GPU models for dynamic workload distribution and scaling to multi-GPU systems to improve sorting efficiency on large datasets.

## Author contributions

**Conceptualization:** Mohammed Alaa Ala'anzy.

**Formal analysis:** Abdulmohsen Algarni.

**Funding acquisition:** Abdulmohsen Algarni.

**Investigation:** Abdulmohsen Algarni.

**Methodology:** Mohammed Alaa Ala'anzy.

**Software:** Nurdaulet Tolendi, Baizhan Baubek.

**Supervision:** Mohammed Alaa Ala'anzy, Abdulmohsen Algarni.

**Writing – original draft:** Mohammed Alaa Ala'anzy, Nurdaulet Tolendi, Baizhan Baubek.

**Writing – review & editing:** Mohammed Alaa Ala'anzy, Abdulmohsen Algarni.

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
