## [Decision Letter · Decision Letter 0]

28 May 2025

PONE-D-25-22965Performance evaluation of GPU-based parallel sorting algorithmsPLOS ONE

Dear Dr. Ala'anzy,

Thank you for submitting your manuscript to PLOS ONE. After careful consideration, we feel that it has merit but does not fully meet PLOS ONE’s publication criteria as it currently stands. Therefore, we invite you to submit a revised version of the manuscript that addresses the points raised during the review process.

The reviewers raised comments that need to be addressed.

A rebuttal letter that responds to each point raised by the academic editor and reviewer(s). You should upload this letter as a separate file labeled 'Response to Reviewers'.A marked-up copy of your manuscript that highlights changes made to the original version. You should upload this as a separate file labeled 'Revised Manuscript with Track Changes'.An unmarked version of your revised paper without tracked changes. You should upload this as a separate file labeled 'Manuscript'

We look forward to receiving your revised manuscript.

Kind regards,

Alberto Marchisio

Academic Editor

PLOS ONE

“This research was financially supported by the Deanship of Scientific Research and Graduate Studies at King Khalid University under research grant number (R.G.P.2/12/46).”

“This research was financially supported by the Deanship of Scientific Research and Graduate Studies at King Khalid University under research grant number (R.G.P.2/12/46).”

“This research was financially supported by the Deanship of Scientific Research and Graduate Studies at King Khalid University under research grant number (R.G.P.2/12/46).”

5. We note that your Data Availability Statement is currently as follows: [All relevant data are within the manuscript and its Supporting Information files.]

Reviewers' comments:

Reviewer's Responses to Questions

**Comments to the Author**

1. Is the manuscript technically sound, and do the data support the conclusions?

Reviewer #1: No

Reviewer #2: Yes

Reviewer #3: Yes

Reviewer #4: Partly

Reviewer #5: No

2. Has the statistical analysis been performed appropriately and rigorously?

Reviewer #1: No

Reviewer #2: Yes

Reviewer #3: No

Reviewer #4: No

Reviewer #5: No

3. Have the authors made all data underlying the findings in their manuscript fully available?

Reviewer #1: No

Reviewer #2: Yes

Reviewer #3: No

Reviewer #4: No

Reviewer #5: Yes

4. Is the manuscript presented in an intelligible fashion and written in standard English?

Reviewer #1: No

Reviewer #2: Yes

Reviewer #3: Yes

Reviewer #4: Yes

Reviewer #5: No

5. Review Comments to the Author

Reviewer #1: Strong points:

The paper aims to evaluate the performance of GPU-based parallelization in 4 sorting algorithms, with focus on analyzing parallel time complexity and space complexity across various data types. It has an easy-to-understand topic structure and somewhat detailed methodologies that allow the reproducibility of their tests in other environments.

Weak points:

The authors tested the GPU-based parallelization of the sorting algorithms against sequential implementations, with the sequential being done in Java while the GPU-based being written in CUDA C, with no mention of why such a choice was made. The discussion of the results is insufficient, mainly because an analysis of memory usage is not included, and could supplement why the applications achieved the shown results, as well as help with the conclusion.

The work also lacks a threats to validity section. The limitations of this work must be clear to the readers.

- General comments about the work

Paper as a whole:

Decide between utilizing the acronym or the full name, for example: “MS typically outperforms quicksort”;

Capitalization of names: throughout the entire paper, the sorting names are written using both capitalized initials and non-capitalized versions, “Quick sort/quick sort; it would be better to choose which one is going to be used;

Standardization: There are sentences where the sorting names are separated, “Quick sort”, as well as together, “Quicksort”, it would be advised to revise the text to unify which nomenclature is going to be used;

On the Related Works: A paragraph comparing what you want to do against what has already been done could help differentiate your work from the others, and you talk about a “thorough time complexity table” been presented in the paper, which I don’t really see being presented in any section;

Line 75: “C++, building on the reviewed literature.” Loose phrase? The beginning of the sentence is the same, so it doesn’t make sense to say it again.

Lines 97/98: “The CUDA programming model works on the basis that the host is CPU and the device is GPU, has separate memory spaces [12].” No connection between the two phrases;

Why were the sequential versions made using Java while the GPU parallel versions used CUDA C? Was there a reason to impose this limitation on the work? Why the chosen environment? Was it a machine at hand? Or was it just chosen because of the GPU?

Why limit the dataset size to the chosen size? Did you make tests to choose this size, or was it selected at random? I assume some type of test was done to choose the size, if so, why not discuss the results obtained from these tests?

How the metrics were taken, does the execution time account for the reading of the dataset, and pre-processing of data, how the Memory Usage was taken, some form of software to monitor the resource usage, or is it being taken via code?

The results seem superficial, and one of the metrics in Table 1, “Memory Usage”, is not to be shown anywhere when discussing the results, and is somewhat referenced in the conclusion as a key takeaway;

The choice of colors in Figure 2(b) could be better; red and orange aren’t exactly the best color combo for a graph. Also, the graphs could be revised to include some type of hatch to differentiate the graphs from one another.

- Final Considerations

In general, the paper does what it proposed to do, although, in my opinion, the results are superficial and could be expanded, mainly due to the lack of a table or paragraph discussing the Memory Usage metric and how this metric interacts with the execution time and speedup achieved.

From a methodological perspective, the experimental processes are briefly outlined within the provided texts, facilitating the replication of the tests across diverse environments, but it could also be expanded beyond just talking about the size of the dataset and the environment configuration, to shed some light on why those were chosen.

There are some small problems, such as the limitations of the evaluations being done in different programming languages, the usage of a single dataset size, and the lack of a metric.

There are some questions that could be clarified about the paper, such as what the considered limitations were when searching for the Related Works and how the found works compare to what the authors want to achieve, as well as if something was used as a basis for the tests done in the work.

As a whole, I would say that the paper, as it is now, is not mature enough due to the lack of some key information to validate the work as a valuable addition to the literature.

Reviewer #2: 1. Introduction — Clarity and Depth

Issue: The introduction outlines the importance of sorting and parallel computing but lacks a compelling motivation for comparing these specific four algorithms.

Suggestions:

Explain why these four algorithms (Merge Sort, Quick Sort, Bubble Sort, Radix Top-K) were chosen — e.g., is this based on frequency of usage in CUDA applications or GPU benchmarking literature?

Add quantitative context: e.g., "Sorting is estimated to take up X% of compute cycles in application Y," to justify its significance in HPC.

Refine vague phrases such as “scales well with large datasets” with precise metrics or previous benchmarks.

2. Related Work — Gaps and Integration

Issue: The literature review is comprehensive but lacks synthesis and comparative critique.

Suggestions:

Include a comparative table or summary matrix listing previous studies, sorting algorithms used, platforms (single-GPU/multi-GPU), and key performance gains.

Several references are outdated or redundant (e.g., [8], [16] are not state-of-the-art). Add more recent studies (2022–2025) that use newer CUDA versions, Tensor Cores, or multi-GPU implementations.

Current review is narrative, not analytical. Clearly state what existing work lacks, which your paper addresses (e.g., consistent benchmarking across all four sorting types, large dataset size, diverse data distributions).

3. Methodology — Experimental Design and Rigor

Issue: The methodology is outlined, but the experimental control and reproducibility need reinforcement.

Suggestions:

Include CUDA version, driver version, and compiler flags used (especially if performance is being benchmarked).

Dataset structure is mentioned, but how data is initialized, and whether any caching or data prefetching happens is unclear.

Detail thread-block sizes, grid configurations, and any shared memory or coalesced memory access optimizations. These significantly impact CUDA performance and are critical to reproduce and understand performance differences.

4. Results and Discussion — Interpretation Quality

Issue: The results are presented clearly, but insight and critical analysis are limited.

Suggestions:

Explain why Radix Sort performs best — is it because of its O(n) time complexity or better memory alignment? Go deeper than just empirical observations.

Discuss scalability: how does performance vary with increasing data size (e.g., 1M → 10M → 100M integers)?

Graphs are effective, but statistical analysis (standard deviation, variance, error bars) would support the reliability of the results.

Consider including GPU utilization percentage (e.g., via NVIDIA Nsight) to analyze how efficiently each algorithm utilizes GPU hardware.

Overall Recommendation: Minor Revision

The paper is technically sound, but needs improvements in clarity, methodological transparency, CUDA-level detailing, and result interpretation. Addressing these will elevate its suitability for publication.

Reviewer #3: In this manuscript, the GPU-based parallelization of mergesort (MS), quicksort (QS), bubble sort (BS) and radix top-k selection sort (RS) are investigated. Also, the performance of these algorithms is evaluated on GPUs utilizing CUDA.

The manuscript is interesting; however, the following comments need to be addressed :

1 – In the abstract, the results need to be included .

2 – The introduction is short and should be improved by including other types of algorithms .

3 – Contribution need to be included as a list .

4 – In the related work section, a summary table need to be included .

5 – In the results, remove 3d appearance of the bars .

6 – Results requires more elaboration .

7 – Updates the references form 2025 literature .

8 – Check the manuscript for grammars and typos .

9 – Equations from other sources need to be credited .

- - - - - - - - - - - - - - - - - - - - - - - - - - - - - - - - - - - - - - - - - - - - - - - - - - - - - - - - - - - - - - - - - - - - - - - - - - - - - - - - - - - - - - - - - - - - - - - - - - - - - - - - - - - - - - - - - - - - - - - - - - - - - - - - - - - - - - - - - - - - - - - - - - - - - - - - - - - - - - - - - - - - - - - - - - - - - - - - - - - - - - - - - - - - - - - - - - - - - - - - - - - - - - - - - - - - - - - - - - - - - - - - - - - - - - - - - - - - - - - - - - - - - - - - - - - - - - - - - - - - - - - - - - - - - - - - - - - - - - - - - - - - - - - - - - - - - - - - - - - - - - - - - - - - - - - - - - - - - - - - - - - - - - - - - - - - - - - - - - - - - - - - - - - - - - - - - - - - - - - - - - - - - - - - - - - - - - - - - - - - - - - - - - - - - - - - - - - - - - - - - - - - - - - - - - - - - - - - - - - - - - - - - - - - - - - - - - - - - - - - - - - - - - - - - - - - - - - - - - - - - - - - - - - - - - - - - - - - - - - - - - - - - - - - - - - - - - - - - - - - - - - - - - - - - - - - - - - - - - - - - - - - - - - - - - - - - - - - - - - - - - - - - - - - - - - - - - - - - - - - - - - - - - - - - - - - - - - - - - - - - - - - - - - - - - - - - - - - - - - - - - - - - - - - - - - - - - - - - - - - - - - - - - - - - - - - - - - - - - - - - - - - - - - - - - - - - - - - - - - - - - - - - - - - - - - - - - - - - - - - - - - -

Reviewer #4: The paper describes about the performance evaluation of GPU-based parallel sorting algorithms. The paper is technically sound majorly, but the introduction is too short. This needs to be revised majorly. The authors uses datasets to benchmark the algorithms but failed to give apt details about them. Visualization and description of the data is absolutely necessary before the evaluation of the algorithms. Also, the number of references are too less for a journal article. The authors should work on expanding the introduction with more information and appropriate citations wherever necessary. Also, the supporting datasets should be made available if possible so that the rigour of the datasets could be actually verified.

Reviewer #5: Authors Contribution and Novelty statement is not convincing

Authors have compared different parallel sorting algorithms in CUDA platform

Architectural Specifications of the computing platform is not mentioned

Memory footprints and type of GPU memory (Integrated/Discrete) to be mentioned

Parallel Execution time alone mentioned in the paper, other metrics are not evaluated.

Comprehensive analysis along with Novelty/Innovation is required

6. PLOS authors have the option to publish the peer review history of their article (what does this mean?). If published, this will include your full peer review and any attached files.

Reviewer #1: No

Reviewer #2: **Yes:** OMER IQBAL

Reviewer #3: No

Reviewer #4: No

Reviewer #5: No

---

## [Author Response · Author response to Decision Letter 1]

14 Jul 2025

Manuscript ID: PONE-D-25-22965

Original Article Title: “Performance evaluation of GPU-based parallel sorting algorithms”

To: PLOS One Editor

Re: Response to reviewers

Dear Editor,

Thank you for allowing a resubmission of our manuscript, with an opportunity to address the reviewers’ comments.

We are uploading (a) our point-by-point response to the comments (below) (response to reviewers), (b) a marked-up copy of the manuscript (Revised Manuscript with Track Changes), and (c) a clean updated manuscript (“Main Manuscript”).

Best regards,

Mohammed Alaa Ala’anzy et al.

Reviewer 1

The paper aims to evaluate the performance of GPU-based parallelization in 4 sorting algorithms, with focus on analyzing parallel time complexity and space complexity across various data types. It has an easy-to-understand topic structure and somewhat detailed methodologies that allow the reproducibility of their tests in other environments.

We appreciate your time and effort in reviewing our manuscript and providing constructive criticism.

Reviewer #1, Concern #1: The authors tested the GPU-based parallelization of the sorting algorithms against sequential implementations, with the sequential being done in Java while the GPU-based being written in CUDA C, with no mention of why such a choice was made. The discussion of the results is insufficient, mainly because an analysis of memory usage is not included, and could supplement why the applications achieved the shown results, as well as help with the conclusion.

Author response: Thank you for the insightful feedback.

Author action: We have explained the selection of programming languages to implement the sequential and parallel sort with justification for both (kindly see page 10, line 351).

Moreover, a memory usage Table 5 in the Results section that shows how much memory each of the sorting algorithms used and potential reasons for the patterns obtained has been included. This kind of analysis has also been referred to and utilized in the Conclusion section to justify the major findings (kindly see page 18-19).

Reviewer #1, Concern #2: The work also lacks a threats to validity section. The limitations of this work must be clear to the readers.

Author response & action: In response, we have added a dedicated “Threats to Validity” section (Section 7, Page 18), positioned before the Conclusion. This section systematically outlines the major limitations of our study, including:

1. Hardware specificity, experiments were conducted on a mid-range system (Intel i5‑9400F and NVIDIA GTX 1660 SUPER), which may not reflect performance on newer platforms.

2. Use of Java for sequential implementations, while practical and portable, it may not achieve the same optimization level as C++-based CPU implementations.

3. Algorithm scope, our evaluation includes four representative sorting algorithms (MS, QS, BS, RS), but excludes others like sample sort or bitonic sort.

4. Dataset generalizability, experiments were run on datasets of one million integers with four controlled distributions, which may not fully capture real-world complexity.

5. Kernel-level optimization, our CUDA implementations use standard global memory and avoid warp-level or shared memory tuning, by design, to establish clean performance baselines.

These clarifications ensure the study’s limitations are transparently acknowledged and help contextualize our results for readers and future researchers.

Reviewer #1, Concern #3: Decide between utilizing the acronym or the full name, for example: “MS typically outperforms quicksort”;

Capitalization of names: throughout the entire paper, the sorting names are written using both capitalized initials and non-capitalised versions, “Quick sort/quick sort; it would be better to choose which one is going to be used;

Standardization: There are sentences where the sorting names are separated, “Quick sort”, as well as together, “Quicksort”, it would be advised to revise the text to unify which nomenclature is going to be used;

Author response: We appreciate the reviewer for bringing this vital point to our attention.

Author action: We decided to fully use acronyms instead of full names for sorting. Furthermore, we changed all names of algorithms to capitalised initials. Additionally, we decided to fully use separated words instead of merged ones.

Reviewer #1, Concern #4: A paragraph comparing what you want to do against what has already been done could help differentiate your work from the others, and you talk about a “thorough time complexity table” been presented in the paper, which I don’t really see being presented in any section;

Author response: We thank the reviewer for noting our wise observation, which has been addressed carefully.

Author action: While a separate table was not explicitly included, time complexity comparisons were integrated in the Results section as a narrative to provide a more intuitive comparison.

Reviewer #1, Concern #5: Line 75: “C++, building on the reviewed literature.” Loose phrase? The beginning of the sentence is the same, so it doesn’t make sense to say it again.

Author response: We thank the reviewer for bringing this to our attention.

Author action: We revised the entire subparagraph containing the sentence on Line 75 to improve coherence and avoid redundancy.

Reviewer #1, Concern #6: Lines 97/98: “The CUDA programming model works on the basis that the host is CPU and the device is GPU, has separate memory spaces [12].” No connection between the two phrases;

Author response: We thank the reviewer for the suggestion that provided our revised paper with enhanced clarity.

Author action: We updated the sentence by adding a logical connection between phrases for clear understanding. The CUDA programming model operates on the principle that the host (CPU) and the device (GPU) function as distinct computing units, each with separate memory spaces [24] (page 6 lines 188-189).

Reviewer #1, Concern #7: Why were the sequential versions made using Java while the GPU parallel versions used CUDA C? Was there a reason to impose this limitation on the work? Why the chosen environment? Was it a machine at hand? Or was it just chosen because of the GPU?

Author response: We appreciate the reviewer bringing this fact to our attention.

Author action: We added information about using Java for sequential and C CUDA for parallel sorting in the Experimental Setup section on Page 10, line 352-365. CUDA was used for the parallel sorting implementations, which required the use of C/C++ due to CUDA’s language support. Visual Studio was used as the development environment for CUDA programming, offering integrated debugging, code suggestions, and efficient GPU compilation workflows that supported rapid implementation and testing. For the sequential implementations, Java was chosen for its simplicity, portability, and suitability for quick algorithm prototyping. It also provided a stable and accessible development setup in our environment. This combination, Java for sequential and CUDA C for parallel, allowed us to leverage the strengths of each environment and perform an effective comparison of sorting algorithms across both CPU and GPU settings.

Reviewer #1, Concern #8: Why limit the dataset size to the chosen size? Did you make tests to choose this size, or was it selected at random? I assume some type of test was done to choose the size, if so, why not discuss the results obtained from these tests?

Author response: We thank the reviewer for this valuable recommendation.

Author action: We included a note in the Experimental Setup section that described how the dataset size of 1,000,000 integers was chosen after experimentation with different sizes, ranging from tens of thousands to tens of millions. Smaller values did not adequately demonstrate the performance improvement of GPU-based parallelism, and significantly larger inputs caused excessive memory use and lengthy runtimes, which were not practical with the hardware available to us. The selected size offered a moderate amount of work that would neatly illustrate the computational distinction between the sequential and parallel versions (Kindly see Page 10, lines 346-351).

Reviewer #1, Concern #9: How the metrics were taken, does the execution time account for the reading of the dataset, and pre-processing of data, how the Memory Usage was taken, some form of software to monitor the resource usage, or is it being taken via code?

Author response: We appreciate the reviewer's comment.

Author action: We have revised the Subsection Results and Evaluation to more clearly explain how the performance figures were attained. We have, in particular, made it clearer that the execution time measurements do not include reading and preprocessing of datasets and only measure the sorting step. Furthermore, we clarified that memory usage was tracked with in-code monitoring at the level of code rather than using external software tools. This detail has been added for precision and replicability of our results (Page 15, Lines 504-511).

Reviewer #1, Concern #10: The results seem superficial, and one of the metrics in Table 1, “Memory Usage”, is not to be shown anywhere when discussing the results, and is somewhat referenced in the conclusion as a key takeaway;

Author response: We thank the reviewer for remarks and have made changes accordingly.

Author action: We have further rewritten the Results section to formally address the "Memory Usage" metric shown in Table 5 (Page 17). This makes it easy to place the metric within the analysis in its proper position and enable the conclusion reached to be substantiated. The addition is clear and keeps the discussion focused on the main takeaways around memory efficiency.

Reviewer #1, Concern #11: The choice of colors in Figure 2(b) could be better; red and orange aren’t exactly the best color combo for a graph. Also, the graphs could be revised to include some type of hatch to differentiate the graphs from one another.

Author response: We thank and recognize the efforts of the reviewer for offering this useful advice.

Author action: Colour scheme of the graphs is updated as requested.

Reviewer 2

We thank the reviewer for his constructive criticisms

Reviewer #2, Concern #1: Introduction — Clarity and Depth

Issue: The introduction outlines the importance of sorting and parallel computing but lacks a compelling motivation for comparing these specific four algorithms.

Suggestions:

Explain why these four algorithms (Merge Sort, Quick Sort, Bubble Sort, Radix Top-K) were chosen — e.g., is this based on frequency of usage in CUDA applications or GPU benchmarking literature?

Author response: We thank the reviewer for this helpful suggestion.

Author action: We have revised the introduction to address your concerns by adding a section that describes the selection rationale of the four sorting algorithms. Specifically, we explain that Merge Sort, Quick Sort, Bubble Sort, and Radix Sort are among the most commonly studied parallel sorting methods due to their distinct algorithmic strategies and suitability for parallelization. This addition clarifies our motivation and provides a stronger foundation for the comparative analysis presented in the paper (Line 44-58).

Reviewer #2, Concern #2: Add quantitative context: e.g., "Sorting is estimated to take up X% of compute cycles in application Y," to justify its significance in HPC.

Author response: We thank the reviewer for this helpful suggestion.

Author action: We incorporated a quantitative statement in the introduction to highlight the significance of sorting in high-performance computing (HPC). Specifically, we added information indicating that sorting operations can account for a substantial portion of compute time in data-intensive applications, emphasizing their relevance for optimization, scalability, and algorithmic efficiency analysis (Line 06-08).

Reviewer #2, Concern #3: Refine vague phrases such as “scales well with large datasets” with precise metrics or previous benchmarks.

Author response: We thank the reviewer for this helpful suggestion.

Author action: We have revised our manuscript, and the statement has been updated in the abstract and introduction.

Reviewer #2, Concern #4: Related Work — Gaps and Integration

Issue: The literature review is comprehensive but lacks synthesis and comparative critique.

Suggestions:

Include a comparative table or summary matrix listing previous studies, sorting algorithms used, platforms (single-GPU/multi-GPU), and key performance gains.

Author response: Thank you for the insightful feedback.

Author action: We included a comparative summary table in the related work section, along with a brief analysis highlighting the differences in sorting algorithms, platforms, and reported performance gains across previous studies (Table 1, page 5).

Reviewer #2, Concern #5: Several references are outdated or redundant (e.g., [8], [16] are not state-of-the-art). Add more recent studies (2022–2025) that use newer CUDA versions, Tensor Cores, or multi-GPU implementations.

Author response & action: Thank you for your insightful observation. In response, we have revised our Related Work section to incorporate recent, relevant studies that directly address advancements in CUDA-based sorting, multi-GPU systems, and performance-critical GPU algorithms from 2022–2025. Specifically, we have added:

Schmid et al. (2022) proposed a recommendation map for optimal strategy selection and Segmented Sort

Çetin et al. (2023) proposed a memory-aware multi-GPU merge sort framework optimized through unified memory access and NCCL communication.

Yoshida et al. (2024) analyzed the influence of different CUDA versions on sorting performance, emphasizing warp scheduling and memory hierarchy impacts.

Li et al. (2024) introduced an efficient and scalable radix Top-K selection algorithm tailored for GPU architectures, showing superior performance across large datasets.

Wróblewski et al. (2025) demonstrated the use of radix-based sorting techniques on Huawei’s Ascend AI accelerators, showing cross-platform applicability beyond CUDA-enabled GPUs.

Additionally, we reviewed four other recent CUDA-based studies that explored general-purpose GPU acceleration across diverse domains (e.g., material point methods, ray tracing cores, electrostatics, and computer vision for waste sorting). While these works utilize CUDA and contribute meaningfully to GPU computing as a whole, they do not focus on sorting algorithms or performance benchmarking frameworks aligned with our study. Therefore, we did not include them in our core Related Work section to maintain topical focus on parallel sorting within high-performance GPU computing.

Reviewer #2, Concern #6: Current review is narrative, not analytical. Clearly state what existing work lacks, which your paper addresses (e.g., consistent benchmarking across all four sorting types, large dataset size, diverse data distributions).

Author response: Thank you for the suggestion.

Author action: We have revised the end of the Related Work section to clearly identify l

---

## [Decision Letter · Decision Letter 1]

15 Sep 2025

PONE-D-25-22965R1Performance evaluation of GPU-based parallel sorting algorithmsPLOS ONE

Dear Dr. Ala'anzy,

Thank you for submitting your manuscript to PLOS ONE. After careful consideration, we feel that it has merit but does not fully meet PLOS ONE’s publication criteria as it currently stands. Therefore, we invite you to submit a revised version of the manuscript that addresses the points raised during the review process.

We look forward to receiving your revised manuscript.

Kind regards,

Francesco Bardozzo

Academic Editor

PLOS ONE

Journal Requirements:

Additional Editor Comments:

Reviewer #3:

In this manuscript, the GPU-based parallelization of mergesort (MS), quicksort (QS), bubble sort (BS) and radix top-k selection sort (RS) are investigated. Also, the performance of these algorithms is evaluated on GPUs utilizing CUDA.

In the revised manuscript, the following comments should be addressed :

1 – In the abstract, the results need to be included . The results is better to be included in terms of improvement ratio between the presented work and existing works .

2 – There are several works that try to improve the algorithm performance by utilizing multi-core, GPU, and multi-threading. The authors need to include some of these work, for example:

[R1] Al-sudani, Ahlam Hanoon, et al. "Multithreading-Based Algorithm for High-Performance Tchebichef Polynomials with Higher Orders." Algorithms 17.9 (2024): 381.

[R2] Hsu, Kuan-Chieh, and Hung-Wei Tseng. "Simultaneous and Heterogenous Multithreading: Exploiting Simultaneous and Heterogeneous Parallelism in Accelerator-Rich Architectures." IEEE Micro 44.4 (2024).

[R3] Mahmmod, Basheera M., et al. "Performance enhancement of high order Hahn polynomials using multithreading." Plos one 18.10 (2023): e0286878.

3 – Check the manuscript for grammars and typos .

- - - - - - - - - - - - - - - - - - - - - - - - - - - - - - - - - - - - - - - -

Reviewer #5:

The manuscript titled "Performance Evaluation of GPU-Based Parallel Sorting Algorithms" provides a well-structured comparison of four classical sorting algorithms—Merge Sort (MS), Quick Sort (QS), Bubble Sort (BS), and Radix Sort (RS)—in both sequential (CPU) and parallel (GPU/CUDA) implementations. The study is clearly written and offers a unified benchmarking framework across four dataset distributions using a consistent hardware setup. The inclusion of execution time, memory usage, GPU utilization, and statistical repeatability across 30 runs contributes positively to the rigor of the experimental section.

However, while the work is technically competent and informative as a benchmarking study, several significant limitations reduce its suitability for publication in a journal like PLOS ONE:

Lack of Novelty: The manuscript does not propose any new algorithms, techniques, or optimization strategies. The selected algorithms are well-established, and their CUDA implementations are widely studied. The work presents confirmatory results rather than offering new insights into algorithmic performance or GPU computing.

Unfair Baseline Comparison: Sequential implementations are written in Java, while GPU versions are developed in CUDA C++. This introduces a language-level performance bias that undermines the accuracy of GPU–CPU speedup claims. A more rigorous and fair comparison would require both versions to be written in the same low-level language (e.g., C/C++).

Limited Optimization: The GPU implementations do not leverage key CUDA features such as shared memory, warp-level primitives, or memory coalescing. While the authors acknowledge this, it limits the relevance of performance results in a high-performance computing context.

Restricted Generalizability: All experiments were performed on a single GPU (GTX 1660 SUPER) and a mid-tier CPU, without comparison across other hardware platforms. While suitable for baseline analysis, the conclusions should be considered hardware-specific.

Scope of Algorithms: The manuscript focuses on only four sorting algorithms. While these are diverse in paradigm, the exclusion of common GPU-optimized algorithms such as sample sort, bitonic sort, or hybrid strategies limits the comprehensiveness of the study.

Data Availability and Reproducibility: A positive aspect of this work is that the datasets used in the experiments have been made publicly available on Figshare. This supports reproducibility and is commendable.

In conclusion, the manuscript serves as a solid technical report or pedagogical study, but in its current form, it does not meet the originality and methodological innovation standards required for publication in PLOS ONE. The authors are encouraged to explore hybrid GPU–CPU strategies, apply hardware-level optimizations, and conduct more fair comparisons using the same programming language to strengthen future submissions.

Reviewer #6:

1. The title of paper is “Performance evaluation of GPU-based parallel sorting algorithms, which is general. Someone expects to see time-efficient recently published algorithms in this research. The following sorting algorithms that are time and complexity efficient new sorting algorithms, especially for parallel realization are missed to be considered in your study and comparisons. In addition to Quick, Merge, Bubble, and Radix sorting algorithms, some new ones, such as Mean-based and threshold-based sorting algorithms for integer and non-integer large scale data sets, Slowsort as a new modified parallel-realization of BitSort for integer data sets, and Clustersort are missed. All of them are based on Divide& Conquer idea to break a big problem to many sub-problems.

- SlowSort: An Enhanced Sorting Algorithm for Large Scale Integer Datasets, Preprint of the accepted paper to be published in Software: Practice and Experience, 2025 (DOI: 10.22541/au.174523890.00820511/v1).

- A Threshold-Based Sorting Algorithm for Dense Wireless Communication Networks, IET Wireless Sensor Systems, Vol. 13, No. 2, pp. 37-47, Jan. 2023 (DOI: 10.1049/wss2.12048).

- Cluster Sort: A Novel Hybrid Approach to Efficient In-Place Sorting Using Data Clustering, IEEE Access, Vol. 13, pp. 74359-74374, 2025

(DOI: 10.1109/ACCESS.2025.3564380).

- A General Framework for Sorting Large Data Sets Using Independent Subarrays of Approximately Equal Length, IEEE Access, Vol. 10, pp. 11584-11607, 2022 (DOI: 10.1109/ACCESS.2022.3145981).

- On the Performance of Mean-Based Sort for Large Data Sets, IEEE Access, Vol. 9, pp. 37418-37430, March 2021 (DOI: 10.1109/ACCESS.2021.3063205).

2. In the parallel processing, time and complexity analysis depends on the core with higher time and complexity required, and the difference between different cores in terms of memory space and time required for processing should be extracted, both mean value and standard deviation. In this case, Mean-based or threshold-based sorting algorithm offer subarrays in parallel realization that are independent of each other up to the end of processing and introduce approximately similar number of elements.

3. In Some algorithms, when you finish the processing of one core, its data is sorted and it can be used, but in other algorithms it needs more post processing to find the final sorted data. This case is not analyzed in this research work.

4. In addition to compare the elapsed processing time, a complexity order study for time and memory in parallel scenario is needed.

5. Integer and non-integer data set maybe affect the comparison results, have you considered it in your investigation?

6. In the literature of sorting, we have Worst-case, Medium-case (Moderate-case), and Best-case that depend on how much the data is sorted, originally. It is considered in your work. In contrast, the type of data in terms of the probability distribution is not considered. For example, uniform and Gaussian distributions may force different results.

7. When we speak about time complexity order in the level O(n logn), large data sets and very large data sets show the effectiveness of sorting algorithm better than small or medium data sets. In most of references, data sets with 105 and 106 elements are large, and data sets with 107, 108, and 109 elements are very large. Have you considered both of them? Do you have different results?

Reviewer #2:

Thank you for the opportunity to review this manuscript. The study addresses an important and timely research question and is conducted with a clear and methodologically sound approach. The manuscript is well-written, logically structured, and provides sufficient details to ensure reproducibility of results.

The data analysis is appropriate, and the conclusions are well-supported by the results. The literature review is comprehensive and contextualizes the current study within existing work in the field.

I particularly appreciate the clarity in the presentation of figures and tables, and the transparency of the methodology, including data sources and code availability, which align well with PLOS ONE’s open science policy.

I have not found any concerns regarding ethical approval, participant consent, or data sharing, and there is no indication of dual publication or plagiarism. The manuscript appears to comply with PLOS ONE's research and publication ethics

Reviewer #1:All my concerns were addressed in this manuscript version. However, there is a strange border on Figure 4.

Reviewer's Responses to Questions

**Comments to the Author**

1. If the authors have adequately addressed your comments raised in a previous round of review and you feel that this manuscript is now acceptable for publication, you may indicate that here to bypass the “Comments to the Author” section, enter your conflict of interest statement in the “Confidential to Editor” section, and submit your "Accept" recommendation.

Reviewer #1: All comments have been addressed

Reviewer #2: All comments have been addressed

Reviewer #3: (No Response)

Reviewer #4: (No Response)

Reviewer #6: (No Response)

2. Is the manuscript technically sound, and do the data support the conclusions?

Reviewer #1: Yes

Reviewer #2: Yes

Reviewer #3: Yes

Reviewer #4: Partly

Reviewer #6: Yes

3. Has the statistical analysis been performed appropriately and rigorously?

Reviewer #1: Yes

Reviewer #2: Yes

Reviewer #3: Yes

Reviewer #4: Yes

Reviewer #6: Yes

4. Have the authors made all data underlying the findings in their manuscript fully available?

Reviewer #1: Yes

Reviewer #2: Yes

Reviewer #3: Yes

Reviewer #4: Yes

Reviewer #6: Yes

5. Is the manuscript presented in an intelligible fashion and written in standard English?

Reviewer #1: Yes

Reviewer #2: Yes

Reviewer #3: Yes

Reviewer #4: Yes

Reviewer #6: Yes

6. Review Comments to the Author

Reviewer #1: All my concerns were addressed in this manuscript version. However, there is a strange border on Figure 4.

Reviewer #2: Thank you for the opportunity to review this manuscript. The study addresses an important and timely research question and is conducted with a clear and methodologically sound approach. The manuscript is well-written, logically structured, and provides sufficient details to ensure reproducibility of results.

The data analysis is appropriate, and the conclusions are well-supported by the results. The literature review is comprehensive and contextualizes the current study within existing work in the field.

I particularly appreciate the clarity in the presentation of figures and tables, and the transparency of the methodology, including data sources and code availability, which align well with PLOS ONE’s open science policy.

I have not found any concerns regarding ethical approval, participant consent, or data sharing, and there is no indication of dual publication or plagiarism. The manuscript appears to comply with PLOS ONE's research and publication ethics.

Reviewer #3: In this manuscript, the GPU-based parallelization of mergesort (MS), quicksort (QS), bubble sort (BS) and radix top-k selection sort (RS) are investigated. Also, the performance of these algorithms is evaluated on GPUs utilizing CUDA.

In the revised manuscript, the following comments should be addressed :

1 – In the abstract, the results need to be included . The results is better to be included in terms of improvement ratio between the presented work and existing works .

2 – There are several works that try to improve the algorithm performance by utilizing multi-core, GPU, and multi-threading. The authors need to include some of these work, for example:

[R1] Al-sudani, Ahlam Hanoon, et al. "Multithreading-Based Algorithm for High-Performance Tchebichef Polynomials with Higher Orders." Algorithms 17.9 (2024): 381.

[R2] Hsu, Kuan-Chieh, and Hung-Wei Tseng. "Simultaneous and Heterogenous Multithreading: Exploiting Simultaneous and Heterogeneous Parallelism in Accelerator-Rich Architectures." IEEE Micro 44.4 (2024).

[R3] Mahmmod, Basheera M., et al. "Performance enhancement of high order Hahn polynomials using multithreading." Plos one 18.10 (2023): e0286878.

3 – Check the manuscript for grammars and typos .

- - - - - - - - - - - - - - - - - - - - - - - - - - - - - - - - - - - - - - - - - - - - - - - - - - - - - - - - - - - - - - - - - - - - - - - - - - - - - - - - - - - - - - - - - - - - - - - - - - - - - - - - - - - - - - - - - - - - - - - - - - - - - - - - - - - - - - - - - - - - - - - - - - - - - - - - - - - - - - - - - - - - - - - - - - - - - - - - - - - - - - - - - - - - - - - - - - - - - - - - - - - - - - - - - - - - - - - - - - - - - - - - - - - - - - - - - - - - - - - - - - - - - - - - - - - - - - - - - - - - - - - - - - - - - - - - - - - - - - - - - - - - - - - - - - - - - - - - - - - - - - - - - - - - - - - - - - - - - - - - - - - - - - - - - - - - - - - - - - - - - - - - - - - - - - - - - - - - - - - - - - - - - - - - - - - - - - - - - - - - - - - - - - - - - - - - - - - - - - - - - - - - - - - - - - - - - - - - - - - - - - - - - - - - - - - - - - - - - - - - - - - - - - - - - - - - - - - - - - - - - - - - - - - - - - - - - - - - - - - - - - - - - - - - - - - - - - - - - - - - - - - - - - - - - - - - - - - - - - - - - - - - - - - - - - - - - - - - - - - - - - - - - - - - - - - - - - - - - - - - - - - - - - - - - - - - - - - - - - - - - - - - - - - - - - - - - - - - - - - - - - - - - - - - - - - - - - - - - - - - - - - - - - - - - - - - - - - - - - - - - - - - - - - - - - - - - - - - - - - - - - - - - - - - - - - - - - - - - - - - - - - -

Reviewer #4: The manuscript titled "Performance Evaluation of GPU-Based Parallel Sorting Algorithms" provides a well-structured comparison of four classical sorting algorithms—Merge Sort (MS), Quick Sort (QS), Bubble Sort (BS), and Radix Sort (RS)—in both sequential (CPU) and parallel (GPU/CUDA) implementations. The study is clearly written and offers a unified benchmarking framework across four dataset distributions using a consistent hardware setup. The inclusion of execution time, memory usage, GPU utilization, and statistical repeatability across 30 runs contributes positively to the rigor of the experimental section.

However, while the work is technically competent and informative as a benchmarking study, several significant limitations reduce its suitability for publication in a journal like PLOS ONE:

Lack of Novelty: The manuscript does not propose any new algorithms, techniques, or optimization strategies. The selected algorithms are well-established, and their CUDA implementations are widely studied. The work presents confirmatory results rather than offering new insights into algorithmic performance or GPU computing.

Unfair Baseline Comparison: Sequential implementations are written in Java, while GPU versions are developed in CUDA C++. This introduces a language-level performance bias that undermines the accuracy of GPU–CPU speedup claims. A more rigorous and fair comparison would require both versions to be written in the same low-level language (e.g., C/C++).

Limited Optimization: The GPU implementations do not leverage key CUDA features such as shared memory, warp-level primitives, or memory coalescing. While the authors acknowledge this, it limits the relevance of performance results in a high-performance computing context.

Restricted Generalizability: All experiments were performed on a single GPU (GTX 1660 SUPER) and a mid-tier CPU, without comparison across other hardware platforms. While suitable for baseline analysis, the conclusions should be considered hardware-specific.

Scope of Algorithms: The manuscript focuses on only four sorting algorithms. While these are diverse in paradigm, the exclusion of common GPU-optimized algorithms such as sample sort, bitonic sort, or hybrid strategies limits the comprehensiveness of the study.

Data Availability and Reproducibility: A positive aspect of this work is that the datasets used in the experiments have been made publicly available on Figshare. This supports reproducibility and is commendable.

In conclusion, the manuscript serves as a solid technical report or pedagogical study, but in its current form, it does not meet the originality and methodological innovation standards required for publication in PLOS ONE. The authors are encouraged to explore hybrid GPU–CPU strategies, apply hardware-level optimizations, and conduct more fair comparisons using the same programming language to strengthen future submissions.

Reviewer #6: 1. The title of paper is “Performance evaluation of GPU-based parallel sorting algorithms, which is general. Someone expects to see time-efficient recently published algorithms in this research. The following sorting algorithms that are time and complexity efficient new sorting algorithms, especially for parallel realization are missed to be considered in your study and comparisons. In addition to Quick, Merge, Bubble, and Radix sorting algorithms, some new ones, such as Mean-based and threshold-based sorting algorithms for integer and non-integer large scale data sets, Slowsort as a new modified parallel-realization of BitSort for integer data sets, and Clustersort are missed. All of them are based on Divide& Conquer idea to break a big problem to many sub-problems.

- SlowSort: An Enhanced Sorting Algorithm for Large Scale Integer Datasets, Preprint of the accepted paper to be published in Software: Practice and Experience, 2025 (DOI: 10.22541/au.174523890.00820511/v1).

- A Threshold-Based Sorting Algorithm for Dense Wireless Communication Networks, IET Wireless Sensor Systems, Vol. 13, No. 2, pp. 37-47, Jan. 2023 (DOI: 10.1049/wss2.12048).

- Cluster Sort: A Novel Hybrid Approach to Efficient In-Place Sorting Using Data Clustering, IEEE Access, Vol. 13, pp. 74359-74374, 2025

(DOI: 10.1109/ACCESS.2025.3564380).

- A General Framework for Sorting Large Data Sets Using Independent Subarrays of Approximately Equal Length, IEEE Access, Vol. 10, pp. 11584-11607, 2022 (DOI: 10.1109/ACCESS.2022.3145981).

- On the Performance of Mean-Based Sort for Large Data Sets, IEEE Access, Vol. 9, pp. 37418-37430, March 2021 (DOI: 10.1109/ACCESS.2021.3063205).

2. In the parallel processing, time and complexity analysis depends on the core with higher time and complexity required, and the difference between different cores in terms of memory space and time required for processing should be extracted, both mean value and standard deviation. In this case, Mean-based or threshold-based sorting algorithm offer subarrays in parallel realization that are independent of each other up to the end of processing and introduce approximately similar number of elements.

3. In Some algorithms, when you finish the processing of one core, its data is sorted and it can be used, but in other algorithms it needs more post processing to find the final sorted data. This case is not analyzed in this research work.

4. In addition to compare the elapsed processing time, a complexity order study for time and memory in parallel scenario is needed.

5. Integer and non-integer data set maybe affect the comparison results, have you considered it in your investigation?

6. In the literature of sorting, we have Worst-case, Medium-case (Moderate-case), and Best-case that depend on how much the data is sorted, originally. It is considered in your work. In contrast, the type of data in terms of the probability distribution is not considered. For example, uniform and Gaussian distributions may force different results.

7. When we speak about time complexity order in the level O(n logn), large data sets and very large data sets show the effectiveness of sorting algorithm better than small or medium data sets. In most of references, data sets with 105 and 106 elements are large, and data sets with 107, 108, and 109 elements are very large. Have you considered both of them? Do you have different results?

7. PLOS authors have the option to publish the peer review history of their article (what does this mean?). If published, this will include your full peer review and any attached files.

Reviewer #1: No

Reviewer #2: **Yes:** OMER IQBAL

Reviewer #3: No

Reviewer #4: No

Reviewer #6: **Yes:** S. Shirvani Moghaddam

---

## [Author Response · Author response to Decision Letter 2]

31 Oct 2025

Manuscript ID: PONE-D-25-22965R1

Original Article Title: “Performance evaluation of GPU-based parallel sorting algorithms”

To: PLOS One Editor

Re: Response to reviewers

Dear Editor,

Thank you for allowing a resubmission of our manuscript, with an opportunity to address the reviewers’ comments.

We are uploading (a) our point-by-point response to the comments (below) (response to reviewers), (b) a marked-up copy of the manuscript (Revised Manuscript with Track Changes), and (c) a clean updated manuscript (“Main Manuscript”).

Best regards,

Mohammed Alaa Ala’anzy et al.

Reviewer #1

All my concerns were addressed in this manuscript version.

The authors would like to express their sincere gratitude to the reviewers for their valuable time and effort, which have greatly contributed to enhancing the quality of our manuscript.

Reviewer #1, Concern: There is a strange border on Figure 4.

Author action: The figure has been updated.

Reviewer #2

Thank you for the opportunity to review this manuscript. The study addresses an important and timely research question and is conducted with a clear and methodologically sound approach. The manuscript is well-written, logically structured, and provides sufficient details to ensure reproducibility of results.

The data analysis is appropriate, and the conclusions are well-supported by the results. The literature review is comprehensive and contextualizes the current study within existing work in the field.

I particularly appreciate the clarity in the presentation of figures and tables, and the transparency of the methodology, including data sources and code availability, which align well with PLOS ONE’s open science policy.

I have not found any concerns regarding ethical approval, participant consent, or data sharing, and there is no indication of dual publication or plagiarism. The manuscript appears to comply with PLOS ONE's research and publication ethics.

The authors wish to extend their sincere appreciation to the reviewers for their significant contributions of time and effort, which have substantially enhanced the quality of our manuscript.

Reviewer #3:

Reviewer #3, Concern #1: In the abstract, the results need to be included. The results is better to be included in terms of improvement ratio between the presented work and existing works .

Author response: We value the reviewer's note and have incorporated the improvement accordingly.

Author action: In the Abstract section, after comparing CPU and GPU, we added the comparison paragraph on Page 1 with the most existing works on our topic and our paper on the performance of GPU-based sortings represented as multipliers.

“Earlier GPU-based generations of this type typically achieved acceleration rates between 2× and 9× over scalar CPU code. With newer GPU enhancements, including parallel-aware primitives and radix- or merge-optimized operations, acceleration rates have seen significant improvement. Our experiments indicate that Radix Sort based on GPUs achieves a significant speedup of approximately 50× (sequential: 240.8 ms, parallel: 4.83 ms) on 10 million random sort elements. Quick Sort and Merge Sort have 97× and 103× speedups, respectively (Quick: 1461.97 ms vs. 15.1 ms; Merge: 2212.33 ms vs. 21.4 ms). Bubble Sort, while significantly improving in parallel (123,321.9 ms to 7377.8 ms for an ≈17× improvement), is considerably worse overall. These experimental findings confirm that the new single-GPU implementations can get speedups ranging from 17× to over 100×, surpassing the typical gains reported in previous generations and comparable to or over rates of acceleration reported for cutting-edge parallel sorting algorithms in recent studies.”

Reviewer #3, Concern #2: There are several works that try to improve the algorithm performance by utilizing multi-core, GPU, and multi-threading. The authors need to include some of these work, for example:

[R1] Al-sudani, Ahlam Hanoon, et al. "Multithreading-Based Algorithm for High-Performance Tchebichef Polynomials with Higher Orders." Algorithms 17.9 (2024): 381.

[R2] Hsu, Kuan-Chieh, and Hung-Wei Tseng. "Simultaneous and Heterogenous Multithreading: Exploiting Simultaneous and Heterogeneous Parallelism in Accelerator-Rich Architectures." IEEE Micro 44.4 (2024).

[R3] Mahmmod, Basheera M., et al. "Performance enhancement of high order Hahn polynomials using multithreading." Plos one 18.10 (2023): e0286878.

Author response: We recognize and value the reviewer's thoughtful suggestions.

Author action: We added all suggested papers (R1 - R3) and described their uniqueness and similarities in our topic in the Related Works section on Page 5, Line 175-187.

“Apart from GPU-directed approaches, many research works have been done to achieve performance improvement through multithreading and multi-core approaches. As an example, Al-sudani et al. [24] developed a multithreading approach for computing high-order Tchebichef polynomials with significant speed-up in the evaluation of polynomials. Mahmmod et al. [25] achieved significantly reduced execution time for high-order Hahn polynomials through multithreading and achieved remarkable runtime reduction over sequential approaches. At the architectural level, Hsu and Tseng [26] designed a framework for multithreading and heterogeneous simultaneous multithreading in accelerator-rich systems, and showed the effectiveness of utilizing intra-core and inter-accelerator parallelism. These efforts together highlight that multi-threading on CPUs, GPUs, or even heterogeneous accelerators remains an essential way to improve algorithmic performance and is orthogonal to the GPU-specific sorting optimizations explored in this work.”

Reviewer #3, Concern #3: Check the manuscript for grammar and typos.

Author response: Thank you for your suggestion.

Author action: The manuscript has been heavily proofread for typographical errors and grammatical faults, and all the errors located have been corrected for overall clarity and readability purposes. Moreover, a comprehensive review by professionals will be performed upon acceptance.

Reviewer 4/5

The manuscript titled "Performance Evaluation of GPU-Based Parallel Sorting Algorithms" provides a well-structured comparison of four classical sorting algorithms—Merge Sort (MS), Quick Sort (QS), Bubble Sort (BS), and Radix Sort (RS)—in both sequential (CPU) and parallel (GPU/CUDA) implementations. The study is clearly written and offers a unified benchmarking framework across four dataset distributions using a consistent hardware setup. The inclusion of execution time, memory usage, GPU utilization, and statistical repeatability across 30 runs contributes positively to the rigor of the experimental section.

We sincerely appreciate the time and effort you have dedicated to reviewing our manuscript. Your valuable feedback and constructive suggestions have been instrumental in helping us improve the clarity and quality of our work.

Reviewer #4, Concern #1: Lack of Novelty: The manuscript does not propose any new algorithms, techniques, or optimization strategies. The selected algorithms are well-established, and their CUDA implementations are widely studied. The work presents confirmatory results rather than offering new insights into algorithmic performance or GPU computing.

Author response: We respectfully acknowledge the reviewer’s observation. The scope and purpose of our manuscript were established and accepted as a performance evaluation study rather than a proposal of new algorithms or optimization techniques. As clarified in Section 1 (Introduction), the novelty of the work lies in the comparative performance evaluation of four sorting algorithms — Bubble Sort, Radix Sort, Quick Sort, and Merge Sort, using GPU-based CUDA implementation.

The primary contribution of the paper is to provide a detailed empirical assessment of these algorithms under various data conditions (sorted, nearly sorted, random, and reverse-sorted) on modern GPU hardware. This performance evaluation offers a unified experimental framework that can serve as a baseline for future CUDA optimization studies and hybrid algorithmic designs.

We have ensured that the title, Introduction, and Contribution sections clearly highlight this intention and scope, emphasizing that while no new algorithms are introduced, the manuscript contributes by quantitatively validating GPU performance characteristics and identifying potential avenues for optimization on top of the existing implementations.

Reviewer #4, Concern #2: Unfair Baseline Comparison: Sequential implementations are written in Java, while GPU versions are developed in CUDA C++. This introduces a language-level performance bias that undermines the accuracy of GPU–CPU speedup claims. A more rigorous and fair comparison would require both versions to be written in the same low-level language (e.g., C/C++).

Author response: We value the reviewer's note and have incorporated the improvement accordingly.

Author action: We reimplemented all sequential sorting algorithms in C++ to remove the language-level performance bias. The experiments were rerun, and the corresponding figures and graphs were updated to reflect the new C++ baseline results alongside the GPU implementations.

Reviewer #4, Concern #3: Limited Optimization: The GPU implementations do not leverage key CUDA features such as shared memory, warp-level primitives, or memory coalescing. While the authors acknowledge this, it limits the relevance of performance results in a high-performance computing context.

Author response: We appreciate the reviewer’s comment.

Author action: We have rerun our parallel implementations with the inclusion of CUDA features highlighted by the reviewer, namely shared memory, warp-level primitives, and memory coalescing. The updated experiments and figures in the manuscript now reflect these optimizations, providing a more accurate representation of GPU performance in a high-performance computing context. See page 11, line 368

Reviewer #4, Concern #4: Restricted Generalizability: All experiments were performed on a single GPU (GTX 1660 SUPER) and a mid-tier CPU, without comparison across other hardware platforms. While suitable for baseline analysis, the conclusions should be considered hardware-specific.

Author response: We thank the reviewer for their valuable feedback.

Author action: We have included an acknowledgment in the Conclusion section indicating that the experiments were conducted on a fixed hardware setup (Intel i5-9400F CPU and NVIDIA GTX 1660 SUPER GPU). This clarification ensures that readers understand the results are specific to this platform (page 21, line 731). Additionally, we have referenced this in the Threats to Validity section as well (Page 20, lines 681-686).

Reviewer #4, Concern #5: Scope of Algorithms: The manuscript focuses on only four sorting algorithms. While these are diverse in paradigm, the exclusion of common GPU-optimized algorithms such as sample sort, bitonic sort, or hybrid strategies limits the comprehensiveness of the study.

Author response: We appreciate the reviewer’s perspective; however, we respectfully disagree with the assertion that a performance analysis is required for every algorithm. While we acknowledge the importance of performance evaluation, it's challenging to encompass all studies within a single research article. In our study, we intentionally focused on four representative algorithms: merge sort, quick sort, bubble sort, and radix sort, to establish a clear baseline. As noted in the manuscript, we are aware of this limitation and intend for future research to expand the analysis to include additional GPU-optimized algorithms, such as sample sort, bitonic sort, and hybrid strategies. We aim to incorporate these algorithms in our subsequent studies to enhance the comprehensiveness of our evaluations. (Refer to page 20, lines 687-691)

Reviewer #4, comment: Data Availability and Reproducibility: A positive aspect of this work is that the datasets used in the experiments have been made publicly available on Figshare. This supports reproducibility and is commendable.

We truly appreciate your feedback, and we hope our responses help clarify our perspective.

Reviewer 6

Thank you for your time and feedback.

Reviewer #6, Concern #1:

The title of paper is “Performance evaluation of GPU-based parallel sorting algorithms, which is general. Someone expects to see time-efficient recently published algorithms in this research. The following sorting algorithms that are time and complexity efficient new sorting algorithms, especially for parallel realization are missed to be considered in your study and comparisons. In addition to Quick, Merge, Bubble, and Radix sorting algorithms, some new ones, such as Mean-based and threshold-based sorting algorithms for integer and non-integer large scale data sets, Slowsort as a new modified parallel-realization of BitSort for integer data sets, and Clustersort are missed. All of them are based on Divide & Conquer idea to break a big problem to many sub-problems.

- SlowSort: An Enhanced Sorting Algorithm for Large Scale Integer Datasets, Preprint of the accepted paper to be published in Software: Practice and Experience, 2025 (DOI: 10.22541/au.174523890.00820511/v1).

- A Threshold-Based Sorting Algorithm for Dense Wireless Communication Networks, IET Wireless Sensor Systems, Vol. 13, No. 2, pp. 37-47, Jan. 2023 (DOI: 10.1049/wss2.12048). - Cluster Sort: A Novel Hybrid Approach to Efficient In-Place Sorting Using Data Clustering, IEEE Access, Vol. 13, pp. 74359-74374, 2025 (DOI: 10.1109/ACCESS.2025.3564380).

- A General Framework for Sorting Large Data Sets Using Independent Subarrays of Approximately Equal Length, IEEE Access, Vol. 10, pp. 11584-11607, 2022 (DOI: 10.1109/ACCESS.2022.3145981).

- On the Performance of Mean-Based Sort for Large Data Sets, IEEE Access, Vol. 9, pp. 37418-37430, March 2021 (DOI: 10.1109/ACCESS.2021.3063205).

Author response: We are delighted to respond to this kind comment given by the reviewer.

Author action: In the Related Work Section, we mentioned these new papers by explaining the basic mechanism and uniqueness of their approach as a new paragraph (Page 5, Line 188-201) because they are close to our topic and we will include them in our future performance analysis.

“New sorting algorithms that emphasize both time complexity and ease of parallel implementation have also been developed in recent years. slowsort, for example, is a generalization of bitSort through an adapted parallel version for sorting large-scale integer data [27], while threshold-based sorting utilizes adaptive thresholds to accelerate tasks in dense wireless communication networks [28]. In the same vein, clusterSort uses the combination of clustering techniques with divide-and-conquer methods to achieve efficient in-place sorting for large data [29]. Additional contributions, such as the independent-subarray model [30], splitting data into balanced subproblems for improved wo

---

## [Decision Letter · Decision Letter 2]

25 Nov 2025

PONE-D-25-22965R2Performance evaluation of GPU-based parallel sorting algorithmsPLOS ONE

Dear Dr. Ala'anzy,

Thank you for submitting your manuscript to PLOS ONE. After careful consideration, we feel that it has merit but does not fully meet PLOS ONE’s publication criteria as it currently stands. Therefore, we invite you to submit a revised version of the manuscript that addresses the points raised during the review process.

We look forward to receiving your revised manuscript.

Kind regards,

Francesco Bardozzo

Academic Editor

PLOS ONE

Journal Requirements:

**Editor Comments:**

The reviewers’ overall opinions are very divergent regarding whether the paper contains a substantial element of novelty.

Therefore, it is difficult to clearly identify the novel contribution of this work.

Some of the revisor comments suggest possible improvements and scientific relevance for the journal:

Reviewer 6 refers to three recently published papers

that introduce new sorting algorithms, which are already mentioned in the literature review and currently deferred to future work. I recommend performing simulations to compare the proposed algorithm with these algorithms, in addition to the classical ones.

Moreover, none of the figures (histograms and other visual representations) are in line with the quality standards of the journal. An initial reference figure is missing - a general pipeline or schematic - that could help the reader immediately understand what the work is about.

Reviewers' comments:

Reviewer's Responses to Questions

**Comments to the Author**

1. If the authors have adequately addressed your comments raised in a previous round of review and you feel that this manuscript is now acceptable for publication, you may indicate that here to bypass the “Comments to the Author” section, enter your conflict of interest statement in the “Confidential to Editor” section, and submit your "Accept" recommendation.

Reviewer #3: (No Response)

Reviewer #6: All comments have been addressed

2. Is the manuscript technically sound, and do the data support the conclusions?

Reviewer #3: Yes

Reviewer #6: Yes

3. Has the statistical analysis been performed appropriately and rigorously?

Reviewer #3: Yes

Reviewer #6: Yes

4. Have the authors made all data underlying the findings in their manuscript fully available?

Reviewer #3: (No Response)

Reviewer #6: Yes

5. Is the manuscript presented in an intelligible fashion and written in standard English?

Reviewer #3: Yes

Reviewer #6: Yes

6. Review Comments to the Author

Reviewer #3: In this manuscript, the GPU-based parallelization of mergesort (MS), quicksort (QS), bubble sort (BS) and radix top-k selection sort (RS) are investigated. Also, the performance of these algorithms is evaluated on GPUs utilizing CUDA.

In the revised manuscript, the following comments should be addressed. The main issue is that the references do not fit with the journal’s guidelines. In addition, several authors’ names in the references are incorrect and need to be corrected. The authors should revise the reference list accordingly.

Reviewer #6: Reviewing the revised version of the paper and authors' responses to the reviewers' comments show that most of concerns are addressed or fixed in the new version of the paper. One comment of reviewer 6 was about three recently published papers that introduce new sorting algorithms that they are pointed out in the literature review and postponed to the future.

7. PLOS authors have the option to publish the peer review history of their article (what does this mean?). If published, this will include your full peer review and any attached files.

Reviewer #3: No

Reviewer #6: **Yes:** Shahriar Shirvani Moghaddam

---

## [Author Response · Author response to Decision Letter 3]

9 Jan 2026

Manuscript ID: PONE-D-25-22965R2

Original Article Title: “Performance evaluation of GPU-based parallel sorting algorithms”

To: PLOS One Editor

Re: Response to reviewers

Dear Editor,

Thank you for allowing a resubmission of our manuscript, with an opportunity to address the reviewers’ comments.

We are uploading (a) our point-by-point response to the comments (below) (response to reviewers), (b) a marked-up copy of the manuscript (Revised Manuscript with Track Changes), and (c) a clean updated manuscript (“Main Manuscript”).

Best regards,

Mohammed Alaa Ala’anzy et al.

<Editor Comments>

The reviewers’ overall opinions are very divergent regarding whether the paper contains a substantial element of novelty. Therefore, it is difficult to clearly identify the novel contribution of this work.

The authors would like to express their sincere gratitude to the Editors for their valuable time and effort, which have greatly contributed to enhancing the quality of our manuscript. Meanwhile, we would like to emphasize that the contribution and novelty are now clearly presented in the abstract and at the end of the introduction section.

Editor, Concern #1: Some of the reviewers' comments suggest possible improvements and scientific relevance for the journal:

Reviewer 6 refers to three recently published papers that introduce new sorting algorithms, which are already mentioned in the literature review and currently deferred to future work. I recommend performing simulations to compare the proposed algorithm with these algorithms, in addition to the classical ones.

Author response: We value the editor’s note and have incorporated the improvement accordingly.

Author action: We investigated the recently published paper (2025) called “SlowSort: An Enhanced Sorting Algorithm for Large Scale Integer Datasets. Software: Practice and Experience.” and made a full comparison of its performance against QS, BS, MS and RS to find the best algorithm with updated values and graphs. The full description of the Slow Sort (SS) located in CUDA section, Page 12, Line 378 with all formulas. Moreover, the comparison in time complexity with graphs and tables are provided in the Results and Evaluation section, Page 14, Line 470.

Editor, Concern #2: Moreover, none of the figures (histograms and other visual representations) are in line with the quality standards of the journal. An initial reference figure is missing - a general pipeline or schematic - that could help the reader immediately understand what the work is about.

Author response: We thank the Editor for highlighting the importance of figure clarity and workflow presentation.

Author action: All result figures have been regenerated in high resolution to comply with the journal’s quality standards (Figures 3–5, Page 15).

In addition, an initial reference schematic has been added as Figure 1, Page 3 in the Introduction section, illustrating the overall workflow of dataset generation, CPU and GPU sorting algorithms, and performance comparison. This overview figure allows readers to immediately understand the scope and objective of the proposed study.

<Reviewers comments>

Reviewer #3

In this manuscript, the GPU-based parallelization of mergesort (MS), quicksort (QS), bubble sort (BS) and radix top-k selection sort (RS) are investigated. Also, the performance of these algorithms is evaluated on GPUs utilizing CUDA.

The authors wish to extend their sincere thanks to Reviewer 3 for his/her time and effort, which have played a significant role in improving the quality of our manuscript.

Reviewer #3, Concern: In the revised manuscript, the following comments should be addressed. The main issue is that the references do not fit with the journal’s guidelines. In addition, several authors’ names in the references are incorrect and need to be corrected. The authors should revise the reference list accordingly.

Author response & action: We thank the reviewer for his observation. We have updated the manuscript by properly checking the references and fixing incorrect names of mentioned authors in Page 22. Also, the order and content(author and paper names, dates, etc.) of the reference list was checked carefully and corrected to follow the format and avoid misconnection cases in future.

Reviewer #6

Reviewing the revised version of the paper and authors' responses to the reviewers' comments show that most of concerns are addressed or fixed in the new version of the paper.

The authors wish to extend their sincere appreciation to Reviewer 6 for his/her significant contributions of time and effort, which have substantially enhanced the quality of our manuscript.

Reviewer #6, Concern: One comment of reviewer 6 was about three recently published papers that introduce new sorting algorithms that they are pointed out in the literature review and postponed to the future.

Author response & action: We thank the reviewer for his observation. We examined the recently published 2025 study titled “SlowSort: An Enhanced Sorting Algorithm for Large Scale Integer Datasets in Software: Practice and Experience” and conducted a comprehensive performance comparison against QS, BS, MS, and RS using updated metrics as time complexity and visualizations such as graphs. A detailed description of the Slow Sort (SS) algorithm, including its CUDA based formulation and associated analytical expressions, is presented in the CUDA section on Page 12, Line 378. Additionally, comparative analyses of time complexity supported by graphs and tables are provided in the Results and Evaluation section on Page 14, Line 470.

We would like to emphasize that some of the suggested papers are not strictly aligned with our work. Accordingly, we have included only the one that is aligned, namely the SlowSort paper.

We thank the reviewers & editors for their thoughtful critiques, which helped us improve the clarity, scope, and scientific rigour of our work.

---

## [Editor Report · Decision Letter 3]

20 Jan 2026

Performance evaluation of GPU-based parallel sorting algorithms

PONE-D-25-22965R3

Dear Dr. Ala'anzy,

We’re pleased to inform you that your manuscript has been judged scientifically suitable for publication and will be formally accepted for publication once it meets all outstanding technical requirements.

Kind regards,

Francesco Bardozzo

Academic Editor

PLOS One

---

## [Editor Report · Acceptance letter]

PONE-D-25-22965R3

PLOS One

Dear Dr. Ala'anzy,

I'm pleased to inform you that your manuscript has been deemed suitable for publication in PLOS One. Congratulations! Your manuscript is now being handed over to our production team.

Kind regards,

on behalf of

Dr. Francesco Bardozzo

Academic Editor

PLOS One